# Synthesis and In Vitro Anti-*Toxoplasma gondii* Activity of Novel Thiazolidin-4-one Derivatives

**DOI:** 10.3390/molecules24173029

**Published:** 2019-08-21

**Authors:** Nazar Trotsko, Adrian Bekier, Agata Paneth, Monika Wujec, Katarzyna Dzitko

**Affiliations:** 1Department of Organic Chemistry, Faculty of Pharmacy with Medical Analytics Division, Medical University of Lublin, 4A Chodźki, 20-093 Lublin, Poland; 2Department of Immunoparasitology, Faculty of Biology and Environmental Protection, University of Lodz, Banacha 12/16, 90-237 Lodz, Poland

**Keywords:** thiazolidin-4-ones, anti-*Toxoplasma gondii* activity, cytotoxicity

## Abstract

Recent findings on the biological activity of thiazolidin-4-ones and taking into account the lack of effective drugs used in the treatment of toxoplasmosis, their numerous side effects, as well as the problem of drug resistance of parasites prompted us to look for new agents. We designed and synthesized a series of new thiazolidin-4-one derivatives through a two-step reaction between 4-substituted thiosemicarbazides with hydroxybenzaldehydes followed by the treatment with ethyl bromoacetate; maleic anhydride and dimethyl acetylenedicarboxylate afforded target compounds. The thiazolidin-4-one derivatives were used to assess the inhibition of *Toxoplasma gondii* growth in vitro. All active thiazolidine-4-one derivatives (12 compounds) inhibited *T. gondii* proliferation in vitro much better than used references drugs both sulfadiazine as well as the synergistic effect of sulfadiazine + trimethoprim (weight ratio 5:1). Most active among them derivatives **94** and **95** showed inhibition of proliferation at about 392-fold better than sulfadiazine and 18-fold better than sulfadiazine with trimethoprim. All active compounds (**82**–**88** and **91**–**95**) against *T. gondii* represent values from 1.75 to 15.86 (CC_30_/IC_50_) lower than no cytotoxic value (CC_30_).

## 1. Introduction

*Toxoplasma gondii* belongs to the cosmopolitan species of the parasite that causes toxoplasmosis in all species of mammals and birds. It is estimated that up to 1/3 of the human population are infected by this parasite [1]. The developed humoral and cellular response in immunocompetent persons quickly limits its intense proliferation but does not completely eliminate it from the host, resulting in the long-term presence of tissue cysts mainly located in the central nervous system (CNS), muscles, and eyes. In turn, the long-term presence of a parasite carries the risk of permanent damage to the visual organ and (or) the brain and is also correlated with the occurrence of serious nervous disorders such as schizophrenia, Parkinson’s disease or epilepsy [2,3]. Infection with protozoan is also associated with serious effects on the health of the fetus and people with immune dysfunction. Newborns with congenital toxoplasmosis are characterized by pathological changes within the nervous system, with the effects of congenital *T. gondii* invasion often noticeable after several years and include damage to the eye or CNS, endocrine disorders, or abnormal sexual development [4]. In immunocompromised persons, toxoplasmosis may manifest itself, among others headaches and around the chest, a sense of confusion, expectoration of blood, problems with breathing because immune system’s lack of ability to fight acute phase of infectious [5]. In addition, there is a risk of permanent damage to the visual organ or the brain in these patients. In extreme cases, this disease is fatal. Currently available drugs not only do not completely eliminate parasitic cysts from the infected organism, and moreover, they show serious side effects, which include hematological and neurological disorders, skin and mucous membrane changes, gastrointestinal disorders, bone marrow damages, teratogenicity, and nephrolithiasis [6,7]. There are also cases of multidrug resistance. There is, therefore, a real need for the development of new drugs that are capable of effective treatment of toxoplasmosis but without side effects.

In recent years, the interest in thiazolidine-4-one derivatives has been increased among scientists due to their broad spectrum of biological activities, including antidiabetic, antibacterial, antifungal, anticancer, and anti-inflammatory, confirmed by numerous reviews on the activity and mechanisms of action of thiazolidine-4-ones [8,9,10,11,12,13]. 

In addition, the lack of 100% of the effectiveness of drugs used in the treatment of toxoplasmosis, their numerous side effects, as well as the problem of increasing drug resistance of parasites prompted us to look for new synthetic compounds that could be used in the future to fight this common parasite. Literature report on the antiparasitic activity of thiazolidine-4-one derivatives [14,15,16] was an inspiration to undertake research to obtain effective low-toxicity compounds with activity against *T. gondii*, belonging to the thiazolidin-4-one group.

## 2. Results and Discussion

### 2.1. Rationale

Hitherto, our research group identified thiazolidine-chlorophenylhydrazone hybrids with antiproliferative activity comparable to that of irinotecan [17] and thiazolidine-thiohydantoin and thiazolidine-chlorophenylthiosemicarbazone hybrids with antibacterial activity comparable to that of cefuroxime [18,19]. The results of the latest research work [20,21,22] in the field of inhibitory activity of thiosemicarbazide derivatives against protozoan *T. gondii*, identified that in particular those compounds which in position 4 contain halogen(alkyl)phenyl group are characterized by antiparasitic activity much more advantageous than the commonly used drug sulfadiazine (IC_50_ 25.67–68.18 μM vs. 10,873.71 μM). These results prompted us to start a pilot synthesis of a series of thiazolidin-4-one derivatives containing in their structure the above-mentioned a thiosemicarbazide moiety (shown in red in Figure 1) to confirm (or exclude) their inhibitory potential against *T. gondii* tachyzoites; from our research to date, it appears that, although there are objective indications of anti-toxoplasmic activity for the proposed group of compounds, strengthened by reports by Góes and colleagues [14,15,16], in practice these compounds may be quite significantly different in bioactivity or even inactive.

### 2.2. Chemistry

We designed novel thiazolidin-4-one derivatives in which N4 and C3 atoms of thiosemicarbazide incorporated in the heterocyclic system creating thiazolidine ring. Others N1 and N2 atoms of thiosemicarbazide were as substitutions in position 2 of thiazolidin-4-ones. The target thiazolidine-4-one derivatives were obtained in the two-step procedure starting with appropriate thiosemicarbazide derivatives. In the first step were obtained thiosemicarbazones (**8**–**40**) by the condensation of thiosemicarbazide derivatives (**1**–**7**) with corresponding hydroxybenzaldehydes. The reactions are illustrated in Scheme 1. 

In the second step were obtained 3 series of compounds. The first series were thiazolidin-4-one derivatives (**41**–**73**), which were obtained by the reaction of thiosemicarbazones (**8**–**40**) with ethyl bromoacetate in anhydrous ethanol and presence of anhydrous sodium acetate. The second series were (4-oxothiazolidin-5-ylo)acetic acid derivatives (**74**–**88**), which were synthesized from corresponding thiosemicarbazones by thia-Michael reaction, involving maleic anhydride as Michael acceptor. And the third series was obtained from appropriate thiosemicarbazones and dimethyl acetylenedicarboxylate, similarly as second group, by Michael addition of the sulfur atom to the triple bond and then cyclization to give (4-oxothiazolidin-5-ylidene)acetic acid derivatives (**89**–**104**). The Scheme 2 was presented synthetic route and reaction condition of these groups of compounds.

The structure of new compounds was confirmed by elemental analysis, ^1^H-NMR and ^13^C-NMR spectra.

All protons from the phenyl fragment of new compounds showed signals at δ ~6.82–7.96 ppm. For *para*-substituted derivatives observed two doublets in above ranges with *J* = 8.7–8.8 Hz. Tri-substituted derivatives showed signals as two doublets and doublet of doublets with coupling constant 1.8–2.1 and 7.8–8.7 Hz in above ranges.

In ^1^H-NMR spectra generally, the protons of CH=N group of all new synthesized compounds show singlet signal at δ ~8.15–8.66 ppm.

For the derivatives **41**–**73** signal from protons of CH_2_ group of thiazolidine ring appearing as a singlet at ranges 4.06–4.16 ppm (compounds **41**–**47**, **52**–**63** and **65**–**73**) and as a doublet of doublets at δ ~4.18–4.28 ppm for compounds **48**–**51** and **64**. 

^1^H-NMR spectra showed resonance assigned to the CH group of (4-oxothiazolidin-5-yl)acetic acid derivatives (**74**–**88**) appearing as triplet or doublet of doublets at 4.53–4.68 ppm (ABX spin system) due to the interaction with methylene group of these derivatives. These results agreed with the data of De Aquino et al. for similar structures [16]. 

Protons signals of =CH-COO group of compounds **89**–**104** were visible as a singlet at 6.80–6.89 ppm range.

In the ^13^C-NMR δ values of the carbons of the all thiazolidin-4-one derivatives (**41**–**104**) showed the carbon signals of C=O groups in the range 171.6–174.2 ppm (compounds **41**–**88**) and at 163.7–166.6 ppm for (4-oxothiazolidin-5-ylidene)acetic acid derivatives (**89**–**104**).

The detailed results of ^1^H-NMR and ^13^C-NMR spectra are presented in Section 3 and Appendix A.

### 2.3. Cytotoxicity of the Thiazolidin-4-one Derivatives against L929 Cells

According to the main assumption, compounds that have been tested for activity against *T. gondii* proliferation should have low cytotoxicity or even they should not be toxic to the host cells. Therefore, in the first stage of the study, a cytotoxicity screen against L929 cells was performed. To described cytotoxicity we use CC_50_ value (cytotoxicity concentration 50%), in accordance with applicable rules, defined as the concentration of test samples that causes 50% destruction of cells, but also CC_30_ value was calculated to determine the initial dose of each compound to anti-*Tg* study (data not shown).

Among the 43 compounds tested derivatives **81**, **96**, **97**, **100**–**104** were insoluble under the protocol conditions and, consequently, they were eliminated from further experiments.

The first series of investigated compounds, that are thiazolidin-4-one derivatives (**42**, **49**, **50**, **53**, **54**, **58**, **59**, **65**, **66**, **68**, and **69**), showed high cytotoxicity CC_50_ = 83.54 ± 8.27 − 385.61 ± 25.60 µM (Table 1), besides compound **43**. The value of CC_50_ for compound **43** was 484.97 ± 20.35 µM but the compound **43** with quite satisfactory cytotoxicity cannot alone represented a group of this derivatives. All compounds of the first group were excluded from further anti-*T. gondii* activity study primarily due to the low CC_50_ value which means as cytotoxicity.

The second series of compounds that are (4-oxothiazolidin-5-ylo)acetic acid derivatives (**74**–**88**) were less cytotoxic than first series. Their CC_50_ were within 424.66 ± 8.96 – ≥ 1000 µM (Table 1).

The third series of compounds that are (4-oxothiazolidin-5-ylidene)acetic acid derivatives (**89**–**95**, **98**, and **99**) (Table 1). In this series, two derivatives with 3-phenyl substituent and two compounds with 3-(4-chlorophenyl) substituent in thiazolidine ring were considered as cytotoxic. The CC_50_ values were 166.89 ± 9.43 − 304.14 ± 11.40 µM. The remaining (4-oxothiazolidin-5-ylidene)acetic acid derivatives showed not high cytotoxicity (CC_50_ = 427.98 ± 21.90 − 492.91 ± 19.89 µM) and were no cytotoxic at the similar level (CC_30_ = 315.97.98 ± 21.91, 404.08 ± 12.59, 440.06 ± 15.78, 448.93 ± 19.48, and 471.94 ± 14.94 µM, data not shown). Data of cytotoxicity of references drugs were also presented in Table 1.

To the next step of our studies forwarded 18 new substances from the second (**75**–**80**, **82**–**88)** and third series (**91**–**95**).

### 2.4. In Vitro Anti-T. gondii Activity of the Thiazolidin-4-ones (**75–99**)

In the next step of our studies, the obtained thiazolidin-4-ones (series 2 and 3) were used to assess the inhibition of *T. gondii* growth in vitro. For this purpose, intracellular parasites (tachyzoites) of RH strain were incubated with different concentrations of the thiazolidin-4-one derivatives. Therefore, that *T. gondii* is an intracellular parasite and host cells should not be destroyed by the cytotoxic concentration of tested compounds, the initial dose of each compound was optimized using CC_30_ value. Inhibition of the parasite growth was monitored by measuring the specific incorporation of [^3^H]uracil in the parasite’s nucleic acids.

Because of absence of detailed information about molecular targets and privileged scaffolds among thiazolidin-4-one derivatives for anti-*T. gondii* activity, we chose group of non-cytotoxic derivatives from series 2 and 3 to study the inhibition of *T. gondii* growth in vitro.

According to results presented in Table 2, the (4-oxothiazolidin-5-yl)acetic acid derivatives (**75**–**80**) with the phenyl substituent at the position 3 of the thiazolidine system were inactive. Change the phenyl substituent to 4-chlorophenyl at the position 3 caused the activity against *T. gondii* proliferation for compounds **82**–**88**. Inhibitory concentration (IC_50_) was at the range 115.92 ± 21.68 – 271.15 ± 24.96 μM. Added another chlorine atom at the phenyl ring in position 3 (**84**→**88**) led to increased activity against *T. gondii* from IC_50_ = 219.93 ± 19.22 μM to IC_50_ = 129.42 ± 14.14 μM (Figure 2).

On the contrary, it seems that the type of substituents (either electron withdrawing or electron donating groups) in the hydroxybenzylidene fragment at the position 2 of the thiazolidine system for compounds **82**–**88** did not have any significant effect on the values of activity against *T. gondii* proliferation (Table 2).

Among the (4-oxothiazolidin-5-ylidene)acetic acid derivatives (**91**–**95**) all compounds showed an activity at the range 27.74 ± 4.27 – 191.21 ± 27.32 μM (Table 2), which was generally better than the activity of their saturated analogues **75**–**80** and **82**–**88** (Figure 2, Figure 3 and Figure 4).

In case of derivatives **91**–**95**, it was not possible to observe the dependence of activity on the substituent’s type at the position 3 (phenyl → 4-chlorophenyl → 2,4-dichlorophenyl) due high cytotoxicity these derivatives (**90**, **98**, and **99**) or insolubility under the protocol conditions (compounds **96**, **97**, and **100**–**104**).

On the other hand, it was the noticed the dependence between substituent’s type in the hydroxybenzylidene fragment at the position 2 of the thiazolidine system and increasing activity against *T. gondii*. Compounds **92** and **93** with electron donating group (methoxy and ethoxy) showed activity with IC_50_ ~180–190 ± 24.50 μM. The absence of the substituent in this fragment (compound **91**) led to increased activity (IC_50_ = 92.88 ± 21.91 μM) and added electron withdrawing group Cl or Br (derivatives **94** and **95**) strongly increased the activity (IC_50_ ~27–28 ± 5.05 μM).

All active (4-oxothiazolidin-5-yl)acetic acid derivatives (**82**–**88**) and (4-oxothiazolidin-5-ylidene)acetic acid derivatives (**91**–**95**) inhibited *T. gondii* proliferation in vitro in concentration lower than used references drugs both sulfadiazine (10,873.71 ± 122.76 μM) as well as synergistic effect of sulfadiazine + trimethoprim (5:1) (510.22 ± 35.87 μM) (Table 2). In addition, compounds **87** and **88** showed inhibition of proliferation at about 90-fold lower IC_50_ value relative to sulfadiazine and more than four times lower relative to sulfadiazine with trimethoprim. Most active among (4-oxothiazolidin-5-ylidene)acetic acid derivatives **94** and **95** showed inhibition of proliferation at about 392-fold better than sulfadiazine and 18-fold better than synergistic effect of sulfadiazine + trimethoprim (5:1).

## 3. Materials and Methods 

### 3.1. Chemistry

All commercial reactants and solvents were purchased from either Alfa Aesar (Kandel, Germany) or Sigma-Aldrich (St. Louis, MO, USA) with the highest purity and used without further purification. The melting points were determined by using Fisher-Johns apparatus (Fisher Scientific, Schwerte, Germany) and are uncorrected. The purity of the compounds was checked by TLC on plates with silica gel Si 60F_254_, produced by Merck Co. (Darmstadt, Germany). The ^1^H-NMR and ^13^C-NMR spectra were recorded by a Bruker Avance 300 MHz instrument using DMSO-d6 as solvent and TMS as an internal standard. Chemical shifts were expressed as δ (ppm). Elemental analyses were performed by AMZ 851 CHX analyzer and the results were within ±0.4% of the theoretical value.

### 3.2. General Method of Synthesis of Thiosemicarbazones (**8**–**40**)

A mixture of 0.01 mol thiosemicarbazide derivatives (**1**–**7**) and 0.01 mol corresponding hydroxybenzaldehydes in 15 mL anhydrous ethanol and presence catalytic amount of glacial acetic acid was refluxed for 0.5**–**2h. The solid obtained after cooling of reaction mixture was filtered off, dried and then recrystallized from acetic acid. Compounds **15**–**18** were recrystallized from isopropanol and compounds **24**–**30** from mixture ethanol-water (1:1).

*1-[(2-Hydroxyphenyl)methylidene]-4-phenyl-3-thiosemicarbazide (**8**):* Yield: 76%, CAS Registry Number: 14938-70-6.

*1-[(3-Hydroxyphenyl)methylidene]-4-phenyl-3-thiosemicarbazide (**9**):* Yield: 78%, CAS Registry Number: 76572-75-3.

*1-[(4-Hydroxyphenyl)methylidene]-4-phenyl-3-thiosemicarbazide (**10**):* Yield: 64%, CAS Registry Number: 76572-74-2.

*>1-[(4-Hydroxy-3-methoxyphenyl)methylidene]-4-phenyl-3-thiosemicarbazide (**11**):* Yield: 80%, CAS Registry Number: 20158-15-0.

*1-[(3-Ethoxy-4-hydroxyphenyl)methylidene]-4-phenyl-3-thiosemicarbazide (**12**):* Yield: 77%, CAS Registry Number: 301350-49-2.

*1-[(3-Chloro-4-hydroxyphenyl)methylidene]-4-phenyl-3-thiosemicarbazide (**13**):* Yield: 70%, CAS Registry Number: 340229-91-6.

*1-[(3-Bromo-4-hydroxyphenyl)methylidene]-4-phenyl-3-thiosemicarbazide (**14**):* Yield: 83%, CAS Registry Number: 1799010-24-4.

*4-(2-Chlorophenyl)-1-[(2-hydroxyphenyl)methylidene]-3-thiosemicarbazide (**15**):* Yield: 80%, mp = 188–190 °C. ^1^H-NMR (DMSO-d_6_) δ (ppm): 6.81–6.90 m, 7.22–7.40 m, 7.55 dd (*J* = 7.8, 1.5 Hz), 7.75 d (*J* = 7.8 Hz), 8.03 d (*J* = 7.8 Hz) (8H, 2-Cl-C_6_**H_4_** and 2-HO-C_6_**H_4_**); 8.49 s (1H, CH=N); 10.01 s (1H, OH); 10.05 s, 11.96 s (2H, 2xNH). Anal. calc. for C_14_H_12_ClN_3_OS (%): C 54.99; H 3.96; N 13.74. Found: C 55.03; H 4.00; N 13.71.

*4-(2-Chlorophenyl)-1-[(3-hydroxyphenyl)methylidene]-3-thiosemicarbazide (**16**):* Yield: 75%, mp = 200–202 ^o^C. ^1^H-NMR (DMSO-d_6_) δ (ppm): 6.82–6.86 m, 7.22–7.41 m, 7.56 dd, 7.77 dd, (8H, 2-Cl-C_6_**H_4_** and 3-HO-C_6_**H_4_**, *J* = 7.8, 1.5 Hz); 8.08 s (1H, CH=N); 9.61 s (1H, OH); 10.08 s, 11.97 s (2H, 2xNH). ^13^C-NMR (DMSO-d_6_) δ (ppm): 114.1; 117.9; 119.4; 127.6; 128.1; 129.8; 129.9; 130.2; 130.8; 135.7; 137.0; 143.8; 158.1; 176.9. Anal. calc. for C_14_H_12_ClN_3_OS (%): C 54.99; H 3.96; N 13.74. Found: C 54.94; H 3.94; N 13.70.

*4-(2-Chlorophenyl)-1-[(4-hydroxyphenyl)methylidene]-3-thiosemicarbazide (**17**):* Yield: 84%, CAS Registry Number: 1097214-18-0.

*4-(2-Chlorophenyl)-1-[(3-ethoxy-4-hydroxyphenyl)methylidene]-3-thiosemicarbazide (**18**):* Yield: 81%, mp = 194–196 ^o^C. ^1^H-NMR (DMSO-d_6_) δ (ppm): 1.34 t (3H, C**H_3_**CH_2_O, *J* = 6.9 Hz); 4.07 q (2H, CH_3_C**H_2_**O, *J* = 6.9 Hz); 6.82 d (*J* = 8.1 Hz), 7.12 dd (*J* = 8.1, 1.8 Hz), 7.30 td (*J* = 7.8, 1.5 Hz), 7.38 td (*J* = 7.8, 1.5 Hz), 7.49 d (*J* = 1.8 Hz), 7.56 dd (*J* = 7.8, 1.5 Hz), 7.85 dd (*J* = 8.1, 1.5 Hz) (7H, 2-Cl-C_6_**H_4_** and 3-C_2_H_5_O-4-HO-C_6_**H_3_**); 8.04 s (1H, CH=N); 9.51 s (1H, OH); 9.98 s, 11.88 s (2H, 2xNH). ^13^C-NMR (DMSO-d_6_) δ (ppm): 15.2; 64.4; 111.1; 115.9; 123.2; 125.7; 127.6; 128.0; 129.6; 129.8; 130.6; 137.0; 144.2; 147.2; 149.9; 176.3. Anal. calc. for C_16_H_16_ClN_3_O_2_S (%): C 54.93; H 4.61; N 12.01. Found: C 54.94; H 4.56; N 12.10.

*4-(3-Chlorophenyl)-1-[(2-hydroxyphenyl)methylidene]-3-thiosemicarbazide (**19**):* Yield: 82%, CAS Registry Number: 67804-97-1.

*4-(3-Chlorophenyl)-1-[(3-hydroxyphenyl)methylidene]-3-thiosemicarbazide (**20**):* Yield: 74%, CAS Registry Number: 2006219-81-2.

*4-(3-Chlorophenyl)-1-[(4-hydroxyphenyl)methylidene]-3-thiosemicarbazide (**21**):* Yield: 79%, CAS Registry Number: 1097214-19-1.

*4-(3-Chlorophenyl)-1-[(4-hydroxy-3-methoxyphenyl)methylidene]-3-thiosemicarbazide (**22**):* Yield: 83%, CAS Registry Number: 70161-62-5.

*4-(3-Chlorophenyl)-1-[(3-ethoxy-4-hydroxyphenyl)methylidene]-3-thiosemicarbazide (**23**):* Yield: 78%, CAS Registry Number: 769153-34-6.

*4-(4-Chlorophenyl)-1-[(2-hydroxyphenyl)methylidene]-3-thiosemicarbazide (**24**):* Yield: 76%, CAS Registry Number: 14121-95-0.

*4-(4-Chlorophenyl)-1-[(3-hydroxyphenyl)methylidene]-3-thiosemicarbazide (**25**):* Yield: 71%, CAS Registry Number: 93535-31-0.

*4-(4-Chlorophenyl)-1-[(4-hydroxyphenyl)methylidene]-3-thiosemicarbazide (**26**):* Yield: 77%, CAS Registry Number: 16434-23-4.

*4-(4-Chlorophenyl)-1-[(4-hydroxy-3-methoxyphenyl)methylidene]-3-thiosemicarbazide (**27**):* Yield: 82%, CAS Registry Number: 16434-24-5.

*4-(4-Chlorophenyl)-1-[(3-ethoxy-4-hydroxyphenyl)methylidene]-3-thiosemicarbazide (**28**):* Yield: 73%, mp = 182–184 °C. ^1^H-NMR (DMSO-d_6_) δ (ppm): 1.35 t (3H, C**H_3_**CH_2_O, *J* = 7.2 Hz); 4.09 q (2H, CH_3_C**H_2_**O, *J* = 7.2 Hz); 6.83 d, 7.19 dd, 7.50 d (3H, 3-C_2_H_5_O-4-HO-C_6_**H_3_**, *J* = 8.1, 1.8 Hz); 7.42 d, 7.62 d (4H, 4-Cl-C_6_**H_4_**, *J* = 8.7 Hz); 8.05 s (1H, CH=N); 9.48 s (1H, OH); 10.02 s, 11.76 s (2H, 2xNH). ^13^C-NMR (DMSO-d_6_) δ (ppm): 15.2; 64.5; 112.0; 115.9; 123.1; 125.7; 128.1; 128.4; 129.7; 138.7; 144.5; 147.6; 149.9; 175.9. Anal. calc. for C_16_H_16_ClN_3_O_2_S (%): C 54.93; H 4.61; N 12.01. Found: C 54.84; H 4.55; N 12.03.

*1-[(3-Chloro-4-hydroxyphenyl)-4-(4-chlorophenyl)methylidene]-3-thiosemicarbazide (**29**):* Yield: 80%, mp = 179–181 °C. ^1^H-NMR (DMSO-d_6_) δ (ppm): 6.99 d, 7.55 dd, 8.09 d (3H, 3-Cl-4-HO-C_6_**H_3_**, *J* = 8.4, 2.1 Hz); 7.43 d, 7.59 d (4H, 4-Cl-C_6_**H_4_**, *J* = 9.0 Hz); 8.04 s (1H, CH=N); 10.14 s (1H, OH); 10.72 s, 11.82 s (2H, 2xNH). ^13^C-NMR (DMSO-d_6_) δ (ppm): 116.9; 121.1; 126.7; 128.3; 128.4; 128.7; 129.1; 129.8; 138.6; 142.7; 155.3; 176.2. Anal. calc. for C_14_H_11_Cl_2_N_3_OS (%): C 49.42; H 3.26; N 12.35. Found: C 49.50; H 3.25; N 12.16.

*1-[(3-Bromo-4-hydroxyphenyl)-4-(4-chlorophenyl)methylidene]-3-thiosemicarbazide (**30**):* Yield: 81%, mp = 188–190 °C. ^1^H-NMR (DMSO-d_6_) δ (ppm): 6.98 d (*J* = 8.4 Hz), 7.42 d (*J* = 8.7 Hz), 7.55–7.61 m, 8.21 d (*J* = 1.8 Hz) (7H, 3-Br-4-HO-C_6_**H_3_** and 4-Cl-C_6_**H_4_**); 8.04 s (1H, CH=N); 10.14 s (1H, OH); 10.78 s, 11.81 s (2H, 2xNH). Anal. calc. for C_14_H_11_BrClN_3_OS (%): C 43.71; H 2.88; N 10.92. Found: C 43.69; H 2.81; N 10.94.

*4-(2,4-Dichlorophenyl)-1-[(4-hydroxy-3-methoxyphenyl)methylidene]-3-thiosemicarbazide (**31**):* Yield: 84%, CAS Registry Number: 70161-69-2.

*4-(4-Bromophenyl)-1-[(3-hydroxyphenyl)methylidene]-3-thiosemicarbazide (**32**):* Yield: 81%, mp = 201–203 °C. ^1^H-NMR (DMSO-d_6_) δ (ppm): 6.82–6.86 m, 7.20–7.30 m (4H, 3-HO-C_6_**H_4_**); 7.55 dd (4H, 4-Br-C_6_H_4_, *J* = 9.9, 8.7 Hz); 8.08 s (1H, CH=N); 9.57 s (1H, OH); 10.11 s, 11.85 s (2H, 2xNH). ^13^C-NMR (DMSO-d_6_) δ (ppm): 114.3; 117.8; 118.0; 119.4; 128.2; 130.1; 131.3; 135.7; 139.0; 144.0; 158.1; 176.3. Anal. calc. for C_14_H_12_BrN_3_OS (%): C 48.01; H 3.45; N 12.00. Found: C 48.09; H 3.41; N 11.94.

*4-(4-Bromophenyl)-1-[(4-hydroxyphenyl)methylidene]-3-thiosemicarbazide (**33**):* Yield: 85%, mp = 200–202 °C. ^1^H-NMR (DMSO-d_6_) δ (ppm): 6.81 d, 7.73 d (4H, 4-HO-C_6_**H_4_**, *J* = 8.7 Hz); 7.56 dd (4H, 4-Br-C_6_H_4_, *J* = 16.2, 9.0 Hz); 8.07 s (1H, CH=N); 9.94 s (1H, OH); 10.01 s, 11.74 s (2H, 2xNH). ^13^C-NMR (DMSO-d_6_) δ (ppm): 116.0; 117.8; 125.3; 128.0; 130.0; 131.3; 139.1; 144.1; 160.0; 175.74. Anal. calc. for C_14_H_12_BrN_3_OS (%): C 48.01; H 3.45; N 12.00. Found: C 48.07; H 3.39; N 11.92.

*4-(4-Fluorophenyl)-1-[(2-hydroxyphenyl)methylidene]-3-thiosemicarbazide (**34**):* Yield: 83%, CAS Registry Number: 16113-37-4.

*4-(4-Fluorophenyl)-1-[(3-hydroxyphenyl)methylidene]-3-thiosemicarbazide (**35**):* Yield: 77%, CAS Registry Number: 908817-47-0.

*4-(4-Fluorophenyl)-1-[(4-hydroxyphenyl)methylidene]-3-thiosemicarbazide (**36**):* Yield: 81%, CAS Registry Number: 16113-39-6.

*4-(4-Fluorophenyl)-1-[(4-hydroxy-3-methoxyphenyl)methylidene]-3-thiosemicarbazide (**37**):* Yield: 79%, CAS Registry Number: 1494-02-6.

*1-[(3-Ethoxy-4-hydroxyphenyl)-4-(4-fluorophenyl)methylidene]-3-thiosemicarbazide (**38**):* Yield: 73%, CAS Registry Number: 769148-51-8.

*1-[(3-Chloro-4-hydroxyphenyl)-4-(4-fluorophenyl)methylidene]-3-thiosemicarbazide (**39**):* Yield: 84%, mp = 193–195 °C. ^1^H-NMR (DMSO-d_6_) δ (ppm): 6.99 d (*J* = 8.4 Hz), 7.21 t (*J* = 8.7 Hz), 7.49–7.56 m, 8.10 d (*J* = 2.1 Hz) (7H, 4-F-C_6_**H_4_** and 3-Cl-4-HO-C_6_**H_3_**); 8.03 s (1H, CH=N); 10.11 s (1H, OH); 10.71 s, 11.77 s (2H, 2xNH). ^13^C-NMR (DMSO-d_6_) δ (ppm): 115.0; 115.3; 116.9; 121.1; 126.7; 128.6; 128.9; 129.0; 129.1; 142.4; 155.3; 176.6. Anal. calc. for C_14_H_11_ClFN_3_OS (%): C 51.94; H 3.42; N 12.98. Found: C 51.88; H 3.40; N 12.94.

*1-[(3-Bromo-4-hydroxyphenyl)-4-(4-fluorophenyl)methylidene]-3-thiosemicarbazide (**40**):* Yield: 83%, mp = 190-192^o^C. ^1^H-NMR (DMSO-d_6_) δ (ppm): 6.97 d, 7.20 t, 7.49-7.60 m, 8.21 s (7H, 4-F-C_6_**H_4_** and 3-Br-4-HO-C_6_**H_3_**, *J* = 8.5 Hz); 8.02 s (1H, CH=N); 10.10 s (1H, OH); 10.75 s, 11.74 s (2H, 2xNH). ^13^C-NMR (DMSO-d_6_) δ (ppm): 108.9; 113.1; 113.5; 114.7;125.3; 127.1; 127.8; 129.9; 134.2; 140.5; 154.4; 174.8. Anal. calc. for C_14_H_11_BrFN_3_OS (%): C 45.67; H 3.01; N 11.41. Found: C 45.61; H 3.02; N 11.37.

### 3.3. General Method of Synthesis of Thiazolidin-4-ones (**41**–**73**)

A mixture of 0.003 mol thiosemicarbazones (**8**–**40**), 0.003 mol ethyl bromoacetate and 0.003 mol anhydrous sodium acetate in 5 ml anhydrous ethanol was refluxed for 2h. After cooling of reaction mixture, the solid obtained was filtered off, dried and then recrystallized from glacial acetic acid.

*2-{[(2-Hydroxyphenyl)methylidene]hydrazinylidene}-3-phenyl-1,3-thiazolidin-4-one (**41**):* Yield 75%, CAS Registry Number: 97636-38-9.

*2-{[(3-Hydroxyphenyl)methylidene]hydrazinylidene}-3-phenyl-1,3-thiazolidin-4-one (**42**):* Yield 87%, mp = 235–236 °C. ^1^H-NMR δ (ppm) (DMSO-d_6_): 4.10 s (2H, CH_2_); 6.83–6.87 m, 7.12–7.26 m, 7.37–7.55 m (9H, C_6_**H_5_** and 3-HO-C_6_**H_4_**); 8.23 s (1H, CH=N); 9.67 s (1H, OH). ^13^C-NMR δ (ppm) (DMSO-d_6_): 32.8; 113.9; 118.6; 119.9; 128.8; 129.2; 129.6; 130.3; 135.5; 135.8; 158.0; 158.4; 165.7; 172.5. Anal. calc. for C_16_H_13_N_3_O_2_S (%): C 61.72; H 4.21; N 13.50. Found: C 61.67; H 4.17; N 13.49.

*2-{[(4-Hydroxyphenyl)methylidene]hydrazinylidene}-3-phenyl-1,3-thiazolidin-4-one (**43**):* Yield 99%, CAS Registry Number: 97738-29-9.

*2-{[(4-Hydroxy-3-methoxyphenyl)methylidene]hydrazinylidene}-3-phenyl-1,3-thiazolidin-4-one (**44**):* Yield 96%, CAS Registry Number: 98052-64-3.

*2-{[(3-Ethoxy-4-hydroxyphenyl)methylidene]hydrazinylidene}-3-phenyl-1,3-thiazolidin-4-one (**45**):* Yield 73%, mp = 229–230 °C. ^1^H-NMR δ (ppm) (DMSO-d_6_): 1.35 t (3H, OCH_2_C**H_3_**, *J* = 6.9 Hz); 3.99–4.08 m (4H, C**H_2_** and OC**H_2_**CH_3_); 6.84 d, 7.15 dd, 7.30 d (3H, 3-C_2_H_5_O-4-HO-C_6_**H_3_**, *J* = 8.1, 1.8 Hz); 7.36–7.54 m (5H, C_6_H_5_); 8.16 s (1H, CH=N); 9.59 bs (1H, OH). ^13^C-NMR δ (ppm) (DMSO-d_6_): 15.2; 32.7; 64.3; 112.2; 116.1; 122.8; 126.0; 128.7; 129.1; 129.5; 135.6; 147.4; 150.4; 158.2; 164.1; 172.4. Anal. calc. for C_18_H_17_N_3_O_3_S (%): C 60.83; H 4.82; N 11.82. Found: C 60.80; H 4.77; N 11.81.

*2-{[(3-Chloro-4-hydroxyphenyl)methylidene]hydrazinylidene}-3-phenyl-1,3-thiazolidin-4-one (**46**):* Yield 83%, mp = 215–216 °C. ^1^H-NMR δ (ppm) (DMSO-d_6_): 4.09 s (2H, CH_2_); 7.03 d (*J* = 8.4 Hz), 7.36–7.56 m, 7.70 d (*J* = 2.1 Hz) (8H, C_6_**H_5_** and 3-Cl-4-HO-C_6_**H_3_**); 8.20 s (1H, CH=N); 10.87 s (1H, OH). ^13^C-NMR δ (ppm) (DMSO-d_6_): 32.8; 117.4; 120.7; 126.8; 128.2; 128.7; 129.1; 129.6; 129.7; 135.5; 156.0; 156.9; 165.0; 172.4. Anal. calc. for C_16_H_12_ClN_3_O_2_S (%): C 55.57; H 3.50; N 12.15. Found: C 55.61; H 3.47; N 12.13.

*2-{[(3-Bromo-4-hydroxyphenyl)methylidene]hydrazinylidene}-3-phenyl-1,3-thiazolidin-4-one (**47**):* Yield 88%, mp = 216–218 °C. ^1^H-NMR δ (ppm) (DMSO-d_6_): 4.09 s (2H, CH_2_); 7.01 d (*J* = 8.4 Hz), 7.36–7.60 m, 7.85 d (*J* = 1.8 Hz) (8H, C_6_**H_5_** and 3-Br-4-HO-C_6_**H_3_**); 8.20 s (1H, CH=N); 10.90 s (1H, OH). ^13^C-NMR δ (ppm) (DMSO-d_6_): 32.8; 110.2; 117.0; 127.3; 128.8; 128.9; 129.1; 129.6; 132.8; 135.5; 156.8; 156.9; 165.0; 172.5. Anal. calc. for C_16_H_12_BrN_3_O_2_S (%): C 49.24; H 3.10; N 10.77. Found: C 49.21; H 3.03; N 10.74.

*3-(2-Chlorophenyl)-2-{[(2-hydroxyphenyl)methylidene]hydrazinylidene}-1,3-thiazolidin-4-one (**48**):* Yield 78%, mp = 168–170 °C. ^1^H-NMR δ (ppm) (DMSO-d_6_): 4.28 dd (2H, CH_2_, *J* = 33.6, 17.4 Hz); 6.89–6.95 m, 7.31–7.37 m, 7.53–7.62 m, 7.68–7.72 m (8H, 2-Cl-C_6_**H_4_** and 2-HO-C_6_**H_4_**); 8.56 s (1H, CH=N); 10.79 s (1H, OH). ^13^C-NMR δ (ppm) (DMSO-d_6_): 33.0; 116.8; 118.7; 120.0; 128.9; 130.5; 131.1; 131.5; 131.7; 132.1; 132.9; 133.0; 158.6; 159.9; 163.7; 171.6. Anal. calc. for C_16_H_12_ClN_3_O_2_S (%): C 55.57; H 3.50; N 12.15. Found: C 55.55; H 3.49; N 12.07.

*3-(2-Chlorophenyl)-2-{[(3-hydroxyphenyl)methylidene]hydrazinylidene}-1,3-thiazolidin-4-one (**49**):* Yield 75%, mp = 192–194 °C. ^1^H-NMR δ (ppm) (DMSO-d_6_): 4.21 dd (2H, CH_2_, *J* = 29.7, 17.4 Hz); 6.83–6.87 m, 7.12–7.26 m, 7.51–7.60 m, 7.67–7.70 m (8H, 2-Cl-C_6_**H_4_** and 3-HO-C_6_**H_4_**); 8.23 s (1H, CH=N); 9.69 s (1H, OH). ^13^C-NMR δ (ppm) (DMSO-d_6_): 32.7; 113.9; 118.7; 120.0; 128.9; 130.3; 130.5; 131.5; 131.6; 132.1; 133.1; 135.6; 158.0; 158.9; 164.2; 171.8. Anal. calc. for C_16_H_12_ClN_3_O_2_S (%): C 55.57; H 3.50; N 12.15. Found: C 55.54; H 3.44; N 12.10.

*3-(2-Chlorophenyl)-2-{[(4-hydroxyphenyl)methylidene]hydrazinylidene}-1,3-thiazolidin-4-one (**50**):* Yield 85%, mp = 178–180 °C. ^1^H-NMR δ (ppm) (DMSO-d_6_): 4.18 dd (2H, CH_2_, *J* = 27.6, 17.1 Hz); 6.82 d, 7.51–7.58 m, 7.67–7.70 m (8H, 2-Cl-C_6_**H_4_** and 4-HO-C_6_**H_4_**, *J* = 8.7 Hz); 8.18 s (1H, CH=N); 10.06 s (1H, OH). ^13^C-NMR δ (ppm) (DMSO-d_6_): 32.6; 116.2; 125.4; 128.9; 130.2; 130.5; 131.5; 131.6; 132.1; 133.2; 158.6; 160.6; 162.5; 171.8. Anal. calc. for C_16_H_12_ClN_3_O_2_S (%): C 55.57; H 3.50; N 12.15. Found: C 55.58; H 3.46; N 12.16.

*3-(2-Chlorophenyl)-2-{[(3-ethoxy-4-hydroxyphenyl)methylidene]hydrazinylidene}-1,3-thiazolidin-4-one (**51**):* Yield 81%, mp = 196–198 °C. ^1^H-NMR δ (ppm) (DMSO-d_6_): 1.34 t (3H, OCH_2_C**H_3_**, *J* = 6.9 Hz); 4.02 q (2H, OC**H_2_**CH_3_, *J* = 6.9 Hz); 4.18 dd (2H, CH_2_, *J* = 26.7, 17.4 Hz); 6.83 d, 7.15 dd, 7.30 d (3H, 3-C_2_H_5_O-4-HO-C_6_**H_3_**, *J* = 8.1, 1.8 Hz); 7.51–7.58 m, 7.66–7.70 m (4H, 2-Cl-C_6_H_4_); 8.16 s (1H, CH=N); 9.61 s (1H, OH). ^13^C-NMR δ (ppm) (DMSO-d_6_): 15.1; 32.6; 64.2; 112.1; 116.1; 122.9; 125.8; 128.9; 130.5; 131.5; 131.6; 132.1; 133.1; 147.4; 150.4; 158.7; 162.6; 171.8. Anal. calc. for C_18_H_16_ClN_3_O_3_S (%): C 55.46; H 4.14; N 10.78. Found: C 55.48; H 4.11; N 10.68.

*3-(3-Chlorophenyl)-2-{[(2-hydroxyphenyl)methylidene]hydrazinylidene}-1,3-thiazolidin-4-one (**52**):* Yield 81%, mp = 194–196 °C. ^1^H-NMR δ (ppm) (DMSO-d_6_): 4.15 s (2H, CH_2_); 6.90–7.00 m, 7.31–7.44 m, 7.53–7.59 m (8H, 3-Cl-C_6_**H_4_** and 2-HO-C_6_**H_4_**); 8.59 s (1H, CH=N); 10.82 s (1H, OH). ^13^C-NMR δ (ppm) (DMSO-d_6_): 33.3; 116.8; 118.8; 120.0; 127.8; 128.9; 129.4; 131.1; 131.2; 132.9; 133.6; 136.6; 158.6; 159.7; 165.0; 172.1. Anal. calc. for C_16_H_12_ClN_3_O_2_S (%): C 55.57; H 3.50; N 12.15. Found: C 55.49; H 3.43; N 12.11.

*3-(3-Chlorophenyl)-2-{[(3-hydroxyphenyl)methylidene]hydrazinylidene}-1,3-thiazolidin-4-one (**53**):* Yield 78%, mp = 198–200 °C. ^1^H-NMR δ (ppm) (DMSO-d_6_): 4.08 s (2H, CH_2_); 6.83–6.87 m, 7.12–7.26 m, 7.39–7.42 m, 7.55–7.57 m (8H, 3-Cl-C_6_**H_4_** and 3-HO-C_6_**H_4_**); 8.25 s (1H, CH=N); 9.66 s (1H, OH). ^13^C-NMR δ (ppm) (DMSO-d_6_): 32.9; 113.9; 118.7; 119.9; 127.8; 128.9; 129.3; 130.3; 131.2; 133.5; 135.7; 136.8; 158.0; 158.6; 165.4; 172.3. Anal. calc. for C_16_H_12_ClN_3_O_2_S (%): C 55.57; H 3.50; N 12.15. Found: C 55.51; H 3.51; N 12.13.

*3-(3-Chlorophenyl)-2-{[(4-hydroxyphenyl)methylidene]hydrazinylidene}-1,3-thiazolidin-4-one (**54**):* Yield 83%, mp = 262–264 °C. ^1^H-NMR δ (ppm) (DMSO-d_6_): 4.06 s (2H, CH_2_); 6.82 d, 7.37–7.41 m, 7.54–7.59 m (8H, 3-Cl-C_6_**H_4_** and 4-HO-C_6_**H_4_**, *J* = 8.7 Hz); 8.21 s (1H, CH=N); 10.05 s (1H, OH). ^13^C-NMR δ (ppm) (DMSO-d_6_): 32.8; 116.2; 125.5; 127.8; 128.9; 129.2; 130.1; 131.2; 133.5; 136.9; 158.4; 160.6; 163.7; 172.3. Anal. calc. for C_16_H_12_ClN_3_O_2_S (%): C 55.57; H 3.50; N 12.15. Found: C 55.53; H 3.47; N 12.12.

*3-(3-Chlorophenyl)-2-{[(4-hydroxy-3-methoxyphenyl)methylidene]hydrazinylidene}-1,3-thiazolidin-4-one (**55**):* Yield 81%, mp = 210–212 °C. ^1^H-NMR δ (ppm) (DMSO-d_6_): 3.79 s (3H, OCH_3_); 4.07 s (2H, CH_2_); 6.83 d, 7.17 dd, 7.32 d (3H, 3-CH_3_O-4-HO-C_6_**H_3_**, *J* = 8.4, 1.8 Hz), 7.38–7.41 m, 7.54–7.56 m (4H, 3-Cl-C_6_H_4_); 8.20 s (1H, CH=N); 9.65 s (1H, OH). ^13^C-NMR δ (ppm) (DMSO-d_6_): 32.8; 56.0; 111.0; 116.1; 122.9; 125.9; 127.8; 128.9; 129.2; 131.1; 133.5; 136.9; 148.3; 150.2; 158.5; 163.8; 172.3. Anal. calc. for C_17_H_14_ClN_3_O_3_S (%): C 54.33; H 3.75; N 11.18. Found: C 54.34; H 3.71; N 11.13.

*3-(3-Chlorophenyl)-2-{[(3-ethoxy-4-hydroxyphenyl)methylidene]hydrazinylidene}-1,3-thiazolidin-4-one (**56**):* Yield 78%, mp = 214–216 °C. ^1^H-NMR δ (ppm) (DMSO-d_6_): 1.35 t (3H, OCH_2_C**H_3_**, *J* = 6.9 Hz); 3.99–4.06 m (4H, C**H_2_** and OC**H_2_**CH_3_); 6.84 d, 7.16 dd, 7.31 d (3H, 3-C_2_H_5_O-4-HO-C_6_**H_3_**, *J* = 8.4, 1.8 Hz), 7.37–7.41 m, 7.54–7.56 m (4H, 3-Cl-C_6_H_4_); 8.18 s (1H, CH=N); 9.58 s (1H, OH). ^13^C-NMR δ (ppm) (DMSO-d_6_): 15.2; 32.6; 63.9; 116.1; 122.8; 125.9; 127.8; 128.4; 129.2; 131.1; 133.5; 136.9; 147.4; 150.4; 155.2; 158.5; 162.1; 172.3. Anal. calc. for C_18_H_16_ClN_3_O_3_S (%): C 55.46; H 4.14; N 10.78. Found: C 55.41; H 4.10; N 10.75.

*3-(4-Chlorophenyl)-2-{[(2-hydroxyphenyl)methylidene]hydrazinylidene}-1,3-thiazolidin-4-one (**57**):* Yield 72%, mp = 238–240 °C. ^1^H-NMR δ (ppm) (DMSO-d_6_): 4.16 s (2H, CH_2_); 6.89–6.95 m, 7.31–7.37 m, 7.54 dd (4H, 2-HO-C_6_**H_4_**, *J* = 7.8, 1.8 Hz); 7.46 d, 7.61 d (4H, 4-Cl-C_6_H_4_, *J* = 8.8 Hz); 8.57 s (1H, CH=N); 10.82 s (1H, OH). ^13^C-NMR δ (ppm) (DMSO-d_6_): 33.2; 116.8; 118.8; 120.0; 129.7; 130.7; 131.1; 132.9; 133.9; 134.2; 158.6; 159.6; 165.1; 172.2. Anal. calc. for C_16_H_12_ClN_3_O_2_S (%): C 55.57; H 3.50; N 12.15. Found: C 55.53; H 3.44; N 12.09.

*3-(4-Chlorophenyl)-2-{[(3-hydroxyphenyl)methylidene]hydrazinylidene}-1,3-thiazolidin-4-one (**58**):* Yield 77%, mp = 230–232 °C. ^1^H-NMR δ (ppm) (DMSO-d_6_): 4.09 s (2H, CH_2_); 6.85 dd, 7.12–7.26 m (4H, 3-HO-C_6_**H_4_**, *J* = 7.8, 1.8 Hz); 7.44 d, 7.60 d (4H, 4-Cl-C_6_H_4_, *J* = 8.7 Hz); 8.24 s (1H, CH=N); 9.80 s (1H, OH). ^13^C-NMR δ (ppm) (DMSO-d_6_): 32.9; 113.9; 118.7; 119.9; 129.6; 130.3; 130.7; 133.8; 134.4; 135.7; 158.1; 158.6; 165.5; 172.4. Anal. calc. for C_16_H_12_ClN_3_O_2_S (%): C 55.57; H 3.50; N 12.15. Found: C 55.50; H 3.46; N 12.08.

*3-(4-Chlorophenyl)-2-{[(4-hydroxyphenyl)methylidene]hydrazinylidene}-1,3-thiazolidin-4-one (**59**):* Yield 75%, mp = 280–282 °C. ^1^H-NMR δ (ppm) (DMSO-d_6_): 4.07 s (2H, CH_2_); 6.83 d (*J* = 8.7 Hz), 7.43 d (*J* = 8.8 Hz), 7.56–7.60 m (8H, 4-Cl-C_6_**H_4_** and 4-HO-C_6_**H_4_**); 8.20 s (1H, CH=N); 10.05 s (1H, OH). ^13^C-NMR δ (ppm) (DMSO-d_6_): 32.8; 116.2; 125.5; 129.6; 130.1; 130.7; 133.7; 134.4; 158.3; 160.6; 163.8; 172.3. Anal. calc. for C_16_H_12_ClN_3_O_2_S (%): C 55.57; H 3.50; N 12.15. Found: C 55.56; H 3.51; N 12.11.

*3-(4-Chlorophenyl)-2-{[(4-hydroxy-3-methoxyphenyl)methylidene]hydrazinylidene}-1,3-thiazolidin-4-one (**60**):* Yield 80%, mp = 226–228 °C, ^1^H-NMR δ (ppm) (DMSO-d_6_): 3.79 s (3H, OCH_3_); 4.07 s (2H, CH_2_); 6.83 d, 7.16 dd, 7.31 d (3H, 3-CH_3_O-4-HO-C_6_**H_3_**, *J* = 8.2, 1.8 Hz); 7.43 d, 7.59 d (4H, 4-Cl-C_6_H_4_, *J* = 8.9 Hz); 8.18 s (1H, CH=N); 9.67 s (1H, OH). ^13^C-NMR δ (ppm) (DMSO-d_6_): 32.8; 55.9; 110.8; 116.0; 122.9; 125.9; 129.6; 130.7; 133.7; 134.4; 148.3; 150.1; 158.4; 163.9; 172.3. Anal. calc. for C_17_H_14_ClN_3_O_3_S (%): C 54.33; H 3.75; N 11.18. Found: C 54.29; H 3.74; N 11.16.

*3-(4-Chlorophenyl)-2-{[(3-ethoxy-4-hydroxyphenyl)methylidene]hydrazinylidene}-1,3-thiazolidin-4-one (**61**):* Yield 83%, mp = 218–220 °C, ^1^H-NMR δ (ppm) (DMSO-d_6_): 1.35 t (3H, OCH_2_C**H_3_**, *J* = 6.9 Hz); 3.99–4.07 m (4H, OC**H_2_**CH_3_ and CH_2_); 6.84 d, 7.15 dd, 7.30 d (3H, 3-C_2_H_5_-4-HO-C_6_**H_3_**, *J* = 8.1, 1.8 Hz); 7.43 d, 7.59 d (4H, 4-Cl-C_6_H_4_, *J* = 8.7 Hz); 8.17 s (1H, CH=N); 9.59 s (1H, OH). ^13^C-NMR δ (ppm) (DMSO-d_6_): 15.2; 32.8; 64.3; 112.2; 116.1; 122.8; 125.9; 129.6; 130.7; 133.7; 134.4; 147.4; 150.4; 158.4; 163.9; 172.3. Anal. calc. for C_18_H_16_ClN_3_O_3_S (%): C 55.46; H 4.14; N 10.78. Found: C 55.43; H 4.15; N 10.69.

*2-{[(3-Chloro-4-hydroxyphenyl)methylidene]hydrazinylidene}-3-(4-chlorophenyl)-1,3-thiazolidin-4-one (**62**):* Yield 71%, mp = 231–233 °C, ^1^H-NMR δ (ppm) (DMSO-d_6_): 4.08 s (2H, CH_2_); 7.03 d, 7.55 dd, 7.70 d (3H, 3-Cl-4-HO-C_6_**H_3_**, *J* = 8.5, 2.0 Hz); 7.43 d, 7.59 d (4H, 4-Cl-C_6_H_4_, *J* = 8.8 Hz); 8.21 s (1H, CH=N); 10.86 s (1H, OH). ^13^C-NMR δ (ppm) (DMSO-d_6_): 32.8; 117.4; 120.7; 126.8; 128.3; 129.6; 129.7; 129.9; 130.4; 130.7; 134.4; 157.1; 164.8; 172.3. Anal. calc. for C_16_H_11_Cl_2_N_3_O_2_S (%): C 50.54; H 2.92; N 11.05. Found: C 50.44; H 2.88; N 10.99.

*2-{[(3-Bromo-4-hydroxyphenyl)methylidene]hydrazinylidene}-3-(4-chlorophenyl)-1,3-thiazolidin-4-one (**63**):* Yield 74%, mp = 235–236 °C, ^1^H-NMR δ (ppm) (DMSO-d_6_): 4.08 s (2H, CH_2_); 7.01 d (*J* = 8.4 Hz), 7.43 d (*J* = 8.7 Hz), 7.58-7.66 m, 7.85 d (*J* = 1.8 Hz) (7H, 3-Br-4-HO-C_6_**H_3_** and 4-Cl-C_6_**H_4_**); 8.20 s (1H, CH=N); 10.91 s (1H, OH). ^13^C-NMR δ (ppm) (DMSO-d_6_): 32.8; 110.2; 117.0; 126.8; 127.2; 128.9; 129.6; 130.7; 132.8; 133.7; 134.4; 157.0; 164.8; 172.3. Anal. calc. for C_16_H_11_BrClN_3_O_2_S (%): C 45.25; H 2.61; N 9.89. Found: C 45.23; H 2.59; N 9.90.

*3-(2,4-Dichlorophenyl)-2-{[(4-hydroxy-3-methoxyphenyl)methylidene]hydrazinylidene}-1,3-thiazolidin-4-one (**64**):* Yield 83%, mp = 140–142 °C, ^1^H-NMR δ (ppm) (DMSO-d_6_): 3.79 s (3H, OCH_3_); 4.18 dd (2H, CH_2_, *J* = 32.1, 17.7 Hz); 6.83 d, 7.16 dd, 7.31 d (3H, 3-CH_3_O-4-HO-C_6_**H_3_**, *J* = 8.4, 1.8 Hz), 7.63 d, 7.90 t (3H, 2,4-diCl-C_6_H_3_, *J* = 1.2 Hz); 8.18 s (1H, CH=N); 9.67 s (1H, OH). ^13^C-NMR δ (ppm) (DMSO-d_6_): 32.7; 56.0; 110.9; 116.0; 123.0; 125.8; 129.1; 130.2; 132.3; 132.8; 133.4; 135.4; 148.3; 150.2; 158.9; 162.2; 171.6. Anal. calc. for C_17_H_13_Cl_2_N_3_O_3_S (%): C 49.77; H 3.19; N 10.24. Found: C 49.74; H 3.12; N 10.23.

*3-(4-Bromophenyl)-2-{[(3-hydroxyphenyl)methylidene]hydrazinylidene}-1,3-thiazolidin-4-one (**65**):* Yield 83%, mp = 242–244 °C. ^1^H-NMR δ (ppm) (DMSO-d_6_): 4.09 s (2H, CH_2_); 6.83–6.87 m, 7.12–7.26 m (4H, 3-HO-C_6_**H_4_**); 7.38 d, 7.73 d (4H, 4-Br-C_6_H_4_, *J* = 8.7 Hz); 8.24 s (1H, CH=N); 9.67 s (1H, OH). ^13^C-NMR δ (ppm) (DMSO-d_6_): 32.9; 113.9; 118.7; 119.9; 122.3; 130.3; 131.0; 132.6; 134.8; 135.7; 158.0; 158.6; 165.4; 172.3. Anal. calc. for C_16_H_12_BrN_3_O_2_S (%): C 49.24; H 3.10; N 10.77. Found: C 49.22; H 3.08; N 10.76.

*3-(4-Bromophenyl)-2-{[(4-hydroxyphenyl)methylidene]hydrazinylidene}-1,3-thiazolidin-4-one (**66**):* Yield 74%, mp = 288–290 °C. ^1^H-NMR δ (ppm) (DMSO-d_6_): 4.07 s (2H, CH_2_); 6.82 d, 7.57 d (4H, 4-HO-C_6_**H_4_**, *J* = 8.7 Hz); 7.37 d, 7.72 d (4H, 4-Br-C_6_H_4_, *J* = 8.7 Hz); 8.19 s (1H, CH=N); 10.04 s (1H, OH). ^13^C-NMR δ (ppm) (DMSO-d_6_): 32.8; 116.2; 122.2; 125.5; 130.1; 131.0; 132.6; 134.9; 158.3; 160.6; 163.7; 172.3. Anal. calc. for C_16_H_12_BrN_3_O_2_S (%): C 49.24; H 3.10; N 10.77. Found: C 49.19; H 3.06; N 10.78.

*3-(4-Fluorophenyl)-2-{[(2-hydroxyphenyl)methylidene]hydrazinylidene}-1,3-thiazolidin-4-one (**67**):* Yield 97%, CAS Registry Number: 16113-43-2.

*3-(4-Fluorophenyl)-2-{[(3-hydroxyphenyl)methylidene]hydrazinylidene}-1,3-thiazolidin-4-one (**68**):* Yield 85%, mp = 204–206 °C. ^1^H-NMR δ (ppm) (DMSO-d_6_): 4.09 s (2H, CH_2_); 6.83–6.86 m, 7.12–7.49 m (8H, 4-F-C_6_**H_4_** and 3-HO-C_6_**H_4_**); 8.23 s (1H, CH=N); 9.66 s (1H, OH). ^13^C-NMR δ (ppm) (DMSO-d_6_): 32.8; 113.9; 116.3; 116.7; 118.6; 119.9; 130.3; 131.0; 131.7; 135.8; 158.1; 158.5; 165.7; 172.5. Anal. calc. for C_16_H_12_FN_3_O_2_S (%): C 58.35; H 3.67; N 12.76. Found: C 58.29; H 3.66; N 12.78.

*3-(4-Fluorophenyl)-2-{[(4-hydroxyphenyl)methylidene]hydrazinylidene}-1,3-thiazolidin-4-one (**69**):* Yield 86%, CAS Registry Number: 16113-47-6.

*3-(4-Fluorophenyl)-2-{[(4-hydroxy-3-methoxyphenyl)methylidene]hydrazinylidene}-1,3-thiazolidin-4-one (**70**):* Yield 79%, mp = 236–238 °C. ^1^H-NMR δ (ppm) (DMSO-d_6_): 3.79 s (3H, OCH_3_); 4.07 s (2H, CH_2_); 6.83 d, 7.17 d, 7.32–7.48 m (7H, 4-F-C_6_**H_4_** and 3-CH_3_O-4-HO-C_6_**H_3_**, *J* = 8.2 Hz); 8.18 s (1H, CH=N); 9.65 s (1H, OH). ^13^C-NMR δ (ppm) (DMSO-d_6_): 32.7; 56.0; 111.0; 116.1; 116.3; 116.6; 122.9; 126.0; 130.9; 131.0; 148.3; 150.2; 158.3; 164.1; 172.4. Anal. calc. for C_17_H_14_FN_3_O_3_S (%): C 56.82; H 3.93; N 11.69. Found: C 56.79; H 3.95; N 11.68.

*2-{[(3-Ethoxy-4-hydroxyphenyl)methylidene]hydrazinylidene}-3-(4-fluorophenyl)-1,3-thiazolidin-4-one (**71**):* Yield 69%, mp = 240–242 °C. ^1^H-NMR δ (ppm) (DMSO-d_6_): 1.35 t (3H, OCH_2_C**H_3_**, *J* = 6.9 Hz); 3.99–4.06 m (4H, C**H_2_** and OC**H_2_**CH_3_); 6.84 d, 7.16 d, 7.30–7.47 m (7H, 4-F-C_6_**H_4_** and 3-C_2_H_5_O-4-HO-C_6_**H_3_**, *J* = 8.2 Hz); 8.17 s (1H, CH=N); 9.57 s (1H, OH). ^13^C-NMR δ (ppm) (DMSO-d_6_): 15.2; 32.7; 64.3; 112.3; 116.1; 116.3; 116.6; 122.8; 126.0; 130.9; 131.1; 147.4; 150.4; 158.3; 164.1; 172.5. Anal. calc. for C_18_H_16_FN_3_O_3_S (%): C 57.90; H 4.32; N 11.25. Found: C 57.87; H 4.33; N 11.18.

*2-{[(3-Chloro-4-hydroxyphenyl)methylidene]hydrazinylidene}-3-(4-fluorophenyl)-1,3-thiazolidin-4-one (**72**):* Yield 76%, mp = 222–224 °C. ^1^H-NMR δ (ppm) (DMSO-d_6_): 4.07 s (2H, CH_2_); 7.03 d, 7.55 dd, 7.70 d (3H, 3-Cl-4-HO-C_6_**H_3_**, *J* = 8.4, 1.8 Hz); 7.32–7.47 m (4H, 4-F-C_6_H_4_); 8.21 s (1H, CH=N); 10.86 s (1H, OH). ^13^C-NMR δ (ppm) (DMSO-d_6_): 32.8; 116.3; 116.6; 117.4; 120.7; 126.8; 128.3; 129.7; 130.9; 131.1; 156.0; 157.0; 165.0; 172.5. Anal. calc. for C_16_H_11_ClFN_3_O_2_S (%): C 52.83; H 3.05; N 11.55. Found: C 52.86; H 3.03; N 11.56.

*2-{[(3-Bromo-4-hydroxyphenyl)methylidene]hydrazinylidene}-3-(4-fluorophenyl)-1,3-thiazolidin-4-one (**73**):* Yield 80%, mp = 224–226 °C. ^1^H-NMR δ (ppm) (DMSO-d_6_): 4.07 s (2H, CH_2_); 7.01 d, 7.59 dd, 7.85 d (3H, 3-Br-4-HO-C_6_**H_3_**, *J* = 8.2, 1.8 Hz); 7.32–7.48 m (4H, 4-F-C_6_H_4_); 8.20 s (1H, CH=N); 10.89 s (1H, OH). ^13^C-NMR δ (ppm) (DMSO-d_6_): 32.8; 110.2; 116.3; 116.7; 117.1; 127.3; 128.9; 130.9; 131.1; 131.7; 132.8; 156.9; 165.0; 172.5. Anal. calc. for C_16_H_11_BrFN_3_O_2_S (%): C 47.07; H 2.72; N 10.29. Found: C 46.99; H 2.69; N 10.26.

### 3.4. General Method of Synthesis of (4-oxothiazolidin-5-ylo)Acetic Acid Derivatives (**74**–**88**)

A mixture of 0.0015 mol thiosemicarbazones (**8**–**14** or **24**–**31**), 0.0015 mol maleic anhydride in 4 mL glacial acetic acid was refluxed for 2h. After cooling of reaction mixture, the solid obtained was filtered off, dried and then recrystallized from glacial acetic acid or isopropanol.

*[2-{[(2-Hydroxyphenyl)methylidene]hydrazinylidene}-4-oxo-3-phenyl-1,3-thiazolidin-5-yl]acetic acid (**74**):* Yield 64%, mp = 228–230 °C. ^1^H-NMR δ (ppm) (DMSO-d_6_): 3.15 d (2H, C**H_2_**CH, *J* = 6.0 Hz); 4.64 t (1H, CH_2_C**H**, *J* = 6.0 Hz); 6.89–6.95 m, 7.31–7.57 m (9H, C_6_**H_5_** and 2-HO-C_6_**H_4_**); 8.56 s (1H, CH=N); 10.83 s (1H, OH); 12.88 bs (1H, COOH). ^13^C-NMR δ (ppm) (DMSO-d_6_): 37.1; 43.5; 116.8; 118.8; 120.0; 128.7; 129.3; 129.6; 131.2; 132.9; 135.5; 158.6; 159.6; 164.6; 172.3; 174.0.

*[2-{[(3-Hydroxyphenyl)methylidene]hydrazinylidene}-4-oxo-3-phenyl-1,3-thiazolidin-5-yl]acetic acid (**75**):* Yield 59%, mp = 214–216 °C. ^1^H-NMR δ (ppm) (DMSO-d_6_): 3.11 d (2H, C**H_2_**CH, *J* = 6.0 Hz); 4.55 t (1H, CH_2_C**H**, *J* = 6.0 Hz); 6.84 dd, 7.10–7.26 m, 7.37–7.55 m (9H, C_6_**H_5_** and 3-HO-C_6_**H_4_**, *J* = 8.1, 1.8 Hz); 8.22 s (1H, CH=N); 9.67 s (1H, OH); 12.85 bs (1H, COOH). ^13^C-NMR δ (ppm) (DMSO-d_6_): 37.3; 43.0; 113.7; 118.6; 120.0; 128.7; 129.2; 129.6; 130.3; 135.6; 135.7; 158.0; 158.3; 165.2; 172.3; 174.2.

*[2-{[(4-Hydroxyphenyl)methylidene]hydrazinylidene}-4-oxo-3-phenyl-1,3-thiazolidin-5-yl]acetic acid (**76**):* Yield 77%, mp = 236–238 °C. ^1^H-NMR δ (ppm) (DMSO-d_6_): 3.09 d (2H, C**H_2_**CH, *J* = 6.0 Hz); 4.53 t (1H, CH_2_C**H**, *J* = 6.0 Hz); 6.82 d, 7.36–7.59 m (9H, C_6_**H_5_** and 4-HO-C_6_**H_4_**, *J* = 8.4 Hz); 8.18 s (1H, CH=N); 10.05 s (1H, OH); 12.54 bs (1H, COOH). ^13^C-NMR δ (ppm) (DMSO-d_6_): 37.4; 42.9; 116.2; 125.6; 128.7; 129.5; 130.1; 131.3; 135.7; 158.1; 160.6; 166.6; 172.3; 174.2.

*[2-{[(4-Hydroxy-3-methoxyphenyl)methylidene]hydrazinylidene}-4-oxo-3-phenyl-1,3-thiazolidin-5-yl]acetic acid (**77**):* Yield 82%, CAS Registry Number: 1008533-45-6.

*[2-{[(3-Ethoxy-4-hydroxyphenyl)methylidene]hydrazinylidene}-4-oxo-3-phenyl-1,3-thiazolidin-5-yl]acetic acid (**78**):* Yield 73%, mp = 224–226 °C. ^1^H-NMR δ (ppm) (DMSO-d_6_): 1.35 t (3H, OCH_2_C**H_3_**, *J* = 6.9 Hz); 3.10 d (2H, C**H_2_**CH, *J* = 6.0 Hz); 4.03 q (2H, OC**H_2_**CH_3_, *J* = 6.9 Hz); 4.53 t (1H, CH_2_C**H**, *J* = 6.0 Hz); 6.83 d, 7.14 dd, 7.29 d (3H, 3-C_2_H_5_O-4-HO-C_6_**H_3_**, *J* = 8.4, 1.8 Hz); 7.35–7.55 m (5H, C_6_H_5_); 8.15 s (1H, CH=N); 9.57 s (1H, OH); 12.81 bs (1H, COOH). ^13^C-NMR δ (ppm) (DMSO-d_6_): 15.2; 37.3; 42.9; 64.3; 112.1; 116.1; 122.9; 125.9; 128.7; 129.1; 129.5; 135.8; 147.5; 150.4; 158.3; 163.3; 172.2; 174.2.

*[2-{[(3-Chloro-4-hydroxyphenyl)methylidene]hydrazinylidene}-4-oxo-3-phenyl-1,3-thiazolidin-5-yl]acetic acid (**79**):* Yield 76%, mp = 244–246 °C. ^1^H-NMR δ (ppm) (DMSO-d_6_): 3.11 d (2H, C**H_2_**CH, *J* = 6.0 Hz); 4.55 t (1H, CH_2_C**H**, *J* = 6.0 Hz); 7.03 d, 7.36–7.56 m, 7.69 d (8H, 3-Cl-4-HO-C_6_**H_3_** and C_6_**H_5_**, *J* = 8.4, 1.8 Hz); 8.20 s (1H, CH=N); 10.83 s (1H, OH); 12.66 bs (1H, COOH). ^13^C-NMR δ (ppm) (DMSO-d_6_): 37.3; 42.9; 117.4; 120.7; 126.9; 128.2; 128.7; 129.2; 129.5; 129.7; 135.7; 155.9; 156.9; 164.4; 172.3; 174.2.

*[2-{[(3-Bromo-4-hydroxyphenyl)methylidene]hydrazinylidene}-4-oxo-3-phenyl-1,3-thiazolidin-5-yl]acetic acid (**80**):* Yield 79%, mp = 252–254 °C. ^1^H-NMR δ (ppm) (DMSO-d_6_): 3.11 d (2H, C**H_2_**CH, *J* = 6.0 Hz); 4.54 t (1H, CH_2_C**H**, *J* = 6.0 Hz); 7.00 d, 7.58 dd, 7.84 d (3H, 3-Br-4-HO-C_6_**H_3_**, *J* = 8.4, 2.1 Hz); 7.35–7.55 m (5H, C_6_H_5_); 8.19 s (1H, CH=N); 10.93 s (1H, OH); 12.63 bs (1H, COOH). ^13^C-NMR δ (ppm) (DMSO-d_6_): 37.2; 42.9; 110.2; 117.0; 127.3; 128.7; 128.9; 129.2; 129.6; 132.8; 135.7; 156.7; 156.9; 164.4; 172.3; 174.2.

*[3-(4-Chlorophenyl)-2-{[(2-hydroxyphenyl)methylidene]hydrazinylidene}-4-oxo-1,3-thiazolidin-5-yl]acetic acid (**81**):* Yield 73%, mp = 234–236 °C. ^1^H-NMR δ (ppm) (DMSO-d_6_): 3.16 d (2H, C**H_2_**CH, *J* = 6.0 Hz); 4.63 t (1H, CH_2_C**H**, *J* = 6.0 Hz); 6.89–7.00 m, 7.31–7.37 m, 7.53–7.55 m (4H, 2-HO-C_6_**H_4_**); 7.45 d, 7.63 d (4H, 4-Cl-C_6_H_4_, *J* = 8.7 Hz); 8.57 s (1H, CH=N); 10.80 s (1H, OH); 12.76 bs (1H, COOH). ^13^C-NMR δ (ppm) (DMSO-d_6_): 37.0; 43.6; 116.8; 118.8; 120.0; 129.7; 130.6; 131.1; 133.0; 133.9; 134.3; 158.6; 159.6; 164.4; 172.3; 173.9.

*[3-(4-Chlorophenyl)-2-{[(3-hydroxyphenyl)methylidene]hydrazinylidene}-4-oxo-1,3-thiazolidin-5-yl]acetic acid (**82**):* Yield 68%, mp = 226–228 °C. ^1^H-NMR δ (ppm) (DMSO-d_6_): 3.12 d (2H, C**H_2_**CH, *J* = 6.0 Hz); 4.55 t (1H, CH_2_C**H**, *J* = 6.0 Hz); 6.83–6.87 m, 7.11–7.13 m, 7.20–7.26 m, (4H, 3-HO-C_6_**H_4_**); 7.43 d, 7.61 d (4H, 4-Cl-C_6_H_4_, *J* = 8.7 Hz); 8.23 s (1H, CH=N); 9.66 s (1H, OH); 12.79 s (1H, COOH). ^13^C-NMR δ (ppm) (DMSO-d_6_): 37.3; 43.1; 113.7; 118.7; 120.0; 129.7; 130.4; 130.6; 133.8; 134.4; 135.7; 158.0; 158.5; 164.9; 172.3; 174.1.

*[3-(4-Chlorophenyl)-2-{[(4-hydroxyphenyl)methylidene]hydrazinylidene}-4-oxo-1,3-thiazolidin-5-yl]acetic acid (**83**):* Yield 75%, mp = 234–235 °C. ^1^H-NMR δ (ppm) (DMSO-d_6_): 3.10 d (2H, C**H_2_**CH, *J* = 6.0 Hz); 4.52 t (1H, CH_2_C**H**, *J* = 6.0 Hz); 6.82 d, 7.57 d (4H, 4-HO-C_6_**H_4_**, *J* = 8.7 Hz); 7.42 d, 7.60 d (4H, 4-Cl-C_6_H_4_, *J* = 8.7 Hz); 8.19 s (1H, CH=N); 10.04 s (1H, OH); 12.84 s (1H, COOH). ^13^C-NMR δ (ppm) (DMSO-d_6_): 37.3; 43.0; 116.2; 125.5; 129.6; 130.1; 130.6; 133.7; 134.5; 158.3; 160.6; 163.2; 172.3; 174.0.

*[3-(4-Chlorophenyl)-2-{[(4-hydroxy-3-methoxyphenyl)methylidene]hydrazinylidene}-4-oxo-1,3-thiazolidin-5-yl]acetic acid (**84**):* Yield 71%, mp = 254–256 °C. ^1^H-NMR δ (ppm) (DMSO-d_6_): 3.11 d (2H, C**H_2_**CH, *J* = 6.0 Hz); 3.79 s (3H, OCH_3_); 4.53 t (1H, CH_2_C**H**, *J* = 6.0 Hz); 6.82 d, 7.15 dd, 7.31 d (3H, 4-HO-3-CH_3_O-C_6_**H_3_**, *J* = 8.1, 1.8 Hz); 7.42 d, 7.61 d (4H, 4-Cl-C_6_H_4_, *J* = 8.7 Hz); 8.18 s (1H, CH=N); 9.66 s (1H, OH); 12.71 s (1H, COOH). ^13^C-NMR δ (ppm) (DMSO-d_6_): 37.2; 43.0; 56.0; 110.7; 116.0; 123.1; 125.9; 129.6; 130.6; 133.7; 134.5; 148.3; 150.2; 158.5; 163.1; 172.3; 174.0.

*[(3-(4-Chlorophenyl)-2-{[(3-ethoxy-4-hydroxyphenyl)methylidene]hydrazinylidene}-4-oxo-1,3-thiazolidin-5-yl]acetic acid (**85**):* Yield 69%, mp = 224–226 °C. ^1^H-NMR δ (ppm) (DMSO-d_6_): 1.35 t (3H, OCH_2_C**H_3_**, *J* = 6.9 Hz); 3.10 d (2H, C**H_2_**CH, *J* = 6.0 Hz); 4.03 q (2H, OC**H_2_**CH_3_, *J* = 6.9 Hz); 4.52 t (1H, C**H_2_**CH, *J* = 6.0 Hz); 6.83 d, 7.14 dd, 7.29 d (3H, 3-C_2_H_5_O-4-HO-C_6_**H_3_**, *J* = 8.1, 1.8 Hz); 7.41 d, 7.60 d (4H, 4-Cl-C_6_H_4_, *J* = 8.7 Hz); 8.16 s (1H, CH=N); 9.58 s (1H, OH); 12.80 s (1H, COOH). ^13^C-NMR δ (ppm) (DMSO-d_6_): 15.2; 37.3; 43.0; 64.3; 112.1; 116.1; 123.0; 125.9; 129.6; 130.6; 133.7; 134.5; 147.5; 150.4; 158.5; 163.0; 172.2; 174.0.

*[2-{[(3-Chloro-4-hydroxyphenyl)methylidene]hydrazinylidene}-3-(4-chlorophenyl)-4-oxo-1,3-thiazolidin-5-yl]acetic acid (**86**):* Yield 77%, mp = 256–257 °C. ^1^H-NMR δ (ppm) (DMSO-d_6_): 3.11 d (2H, C**H_2_**CH, *J* = 6.0 Hz); 4.53 t (1H, CH_2_C**H**, *J* = 6.0 Hz); 7.03 d, 7.55 dd, 7.69 d (3H, 3-Cl-4-HO-C_6_**H_3_**, *J* = 8.7, 1.8 Hz); 7.42 d, 7.61 d (4H, 4-Cl-C_6_H_4_, *J* = 8.7 Hz); 8.20 s (1H, CH=N); 10.86 s (1H, OH); 12.82 s (1H, COOH). ^13^C-NMR δ (ppm) (DMSO-d_6_): 37.2; 43.0; 117.4; 120.7; 126.8; 128.3; 129.7; 129.8; 130.6; 133.8; 134.5; 156.0; 157.1; 164.2; 172.3; 174.0.

*[2-{[(3-Bromo-4-hydroxyphenyl)methylidene]hydrazinylidene}-3-(4-chlorophenyl)-4-oxo-1,3-thiazolidin-5-yl]acetic acid (**87**):* Yield 62%, mp = 254–255 °C. ^1^H-NMR δ (ppm) (DMSO-d_6_): 3.11 d (2H, C**H_2_**CH, *J* = 6.0 Hz); 4.53 t (1H, CH_2_C**H**, *J* = 6.0 Hz); 7.01 d, 7.42 d, 7.57-7.62 m, 7.84 d (7H, 3-Br-4-HO-C_6_**H_3_**, *J* = 8.4, 1.8 Hz and Cl-C_6_**H_4_**, *J* = 8.7 Hz); 8.20 s (1H, CH=N); 10.92 s (1H, OH); 12.82 s (1H, COOH). ^13^C-NMR δ (ppm) (DMSO-d_6_): 37.2; 43.0; 110.2; 117.0; 127.2; 128.9; 129.7; 130.6; 132.8; 133.8; 134.5; 156.9; 157.0; 164.2; 172.3; 174.1.

*[3-(2,4-Dichlorophenyl)-2-{[(4-hydroxy-3-methoxyphenyl)methylidene]hydrazinylidene}-4-oxo-1,3-thiazolidin-5-yl]acetic acid (**88**):* Yield 66%, mp = 226–228 °C. ^1^H-NMR δ (ppm) (DMSO-d_6_): 3.12–3.16 m (2H, C**H_2_**CH); 3.79 s (3H, OCH_3_); 4.68 dd (1H, CH_2_C**H**, *J* = 6.9, 4.2 Hz); 6.82 d, 7.13–7.17 m, 7.30 s, 7.54 d, 7.63–7.68 m, 7.89–7.92 m (6H, 4-HO-3-CH_3_O-C_6_**H_3_** and 2,4-diCl-C_6_**H_3_**, *J* = 8.4 Hz); 8.17 s (1H, CH=N); 9.69 s (1H, OH); 12.85 bs (1H, COOH). ^13^C-NMR δ (ppm) (DMSO-d_6_): 37.3; 42.8; 56.0; 110.8; 116.0; 123.2; 125.7; 129.2; 130.2; 132.4; 133.5; 133.6; 135.4; 148.3; 150.3; 159.0; 161.4; 172.2; 173.4.

### 3.5. General Method of Synthesis of (4-oxothiazolidin-5-ylidene)Acetic Acid Derivatives (**89**–**104**)

A mixture of 0.0015 mol thiosemicarbazones (**8**–**14**, **18** or **24**–**31**), 0.0015 mol dimethyl acetylenedicarboxylate in 10 mL methanol was reflux for 1.5h. After cooling of reaction mixture, the solid obtained was filtered off, dried and then recrystallized from glacial acetic acid.

*Methyl [2-{[(2-hydroxyphenyl)methylidene]hydrazinylidene}-4-oxo-3-phenyl-1,3-thiazolidin-5-ylidene]acetate (**89**):* Yield 90%, mp = 242–244 °C. ^1^H-NMR δ (ppm) (DMSO-d_6_): 3.83 s (3H, COOCH_3_); 6.85 s (1H, CH=); 6.91–6.96 m, 7.34–7.40 m, 7.50–7.57 m, 7.68 dd (9H, 2-HO-C_6_**H_4_** and C_6_**H_5_**, *J* = 8.1, 1.8 Hz); 8.65 s (1H, CH=N); 10.53 s (1H, OH). ^13^C-NMR δ (ppm) (DMSO-d_6_): 53.2; 115.7; 116.9; 119.1; 120.2; 128.7; 129.6; 129.7; 129.8; 133.6; 134.6; 141.7; 158.5; 159.6; 160.4; 164.4; 166.5.

*Methyl [2-{[(3-hydroxyphenyl)methylidene]hydrazinylidene}-4-oxo-3-phenyl-1,3-thiazolidin-5-ylidene]acetate (**90**):* Yield 84%, mp = 236–238 °C. ^1^H-NMR δ (ppm) (DMSO-d_6_): 3.83 s (3H, COOCH_3_); 6.82 s (1H, CH=); 6.88–6.91 m, 7.17–7.19 m, 7.25–7.30 m, 7.49–7.58 m (9H, 3-HO-C_6_**H_4_** and C_6_**H_5_**); 8.38 s (1H, CH=N); 9.82 s (1H, OH). ^13^C-NMR δ (ppm) (DMSO-d_6_): 53.1; 113.7; 115.4; 119.4; 120.6; 128.7; 129.5; 129.6; 130.5; 134.7; 135.2; 142.3; 158.1; 160.6; 161.3; 164.7; 166.6.

*Methyl [2-{[(4-hydroxyphenyl)methylidene]hydrazinylidene}-4-oxo-3-phenyl-1,3-thiazolidin-5-ylidene]acetate (**91**):* Yield 81%, mp = 222–224 °C. ^1^H-NMR δ (ppm) (DMSO-d_6_): 3.83 s (3H, COOCH_3_); 6.80 s (1H, CH=); 6.86 d, 7.64 d (4H, 4-HO-C_6_**H_4_**, *J* = 8.7 Hz); 7.47–7.57 m (5H, C_6_H_5_); 8.33 s (1H, CH=N); 10.22 s (1H, OH). ^13^C-NMR δ (ppm) (DMSO-d_6_): 53.0; 115.1; 116.3; 125.0; 128.7; 129.5; 129.6; 130.6; 134.8; 142.5; 159.6; 160.3; 161.2; 164.6; 166.5.

*Methyl [2-{[(4-hydroxy-3-methoxyphenyl)methylidene]hydrazinylidene}-4-oxo-3-phenyl-1,3-thiazolidin-5-ylidene]acetate (**92**):* Yield 79%, mp = 210–212 °C. ^1^H-NMR δ (ppm) (DMSO-d_6_): 3.81 s (3H, OCH_3_); 3.82 s (3H, COOCH_3_); 6.80 s (1H, CH=); 6.87 d, 7.25 dd, 7.35 d (3H, 4-HO-3-CH_3_O-C_6_**H_3_**, *J* = 8.1, 1.8 Hz); 7.49-7.58 m (5H, C_6_H_5_); 8.31 s (1H, CH=N); 9.84 s (1H, OH). ^13^C-NMR δ (ppm) (DMSO-d_6_): 53.1; 56.0; 111.3; 115.1; 116.2; 123.4; 125.4; 128.7; 129.5; 129.6; 134.8; 142.5; 148.3; 150.8; 159.4; 160.6; 164.6; 166.5.

*Methyl [2-{[(3-ethoxy-4-hydroxyphenyl)methylidene]hydrazinylidene}-4-oxo-3-phenyl-1,3-thiazolidin-5-ylidene]acetate (**93**):* Yield 81%, mp = 190–192 °C. ^1^H-NMR δ (ppm) (DMSO-d_6_): 1.36 t (3H, OCH_2_C**H_3_**, *J* = 6.9 Hz); 3.82 s (3H, COOCH_3_); 4.06 q (2H, OC**H_2_**CH_3_, *J* = 6.9 Hz); 6.80 s (1H, CH=); 6.88 d, 7.25 dd, 7.33 d (3H, 3-C_2_H_5_O-4-HO-C_6_**H_3_**, *J* = 8.1, 1.8 Hz); 7.47–7.57 m (5H, C_6_H_5_); 8.30 s (1H, CH=N); 9.74 s (1H, OH). ^13^C-NMR δ (ppm) (DMSO-d_6_): 15.2; 53.1; 64.4; 112.8; 115.1; 116.3; 123.3; 125.4; 128.7; 129.5; 129.6; 137.8; 142.4; 147.5; 151.0; 159.3; 160.6; 164.6; 166.5.

*Methyl [2-{[(3-chloro-4-hydroxyphenyl)methylidene]hydrazinylidene}-4-oxo-3-phenyl-1,3-thiazolidin-5-ylidene]acetate (**94**):* Yield 83%, mp = 238–240 °C. ^1^H-NMR δ (ppm) (DMSO-d_6_): 3.83 s (3H, COOCH_3_); 6.81 s (1H, CH=); 7.07 d, 7.63 dd, 7.74 d (3H, 3-Cl-4-HO-C_6_**H_3_**, *J* = 8.4, 2.0 Hz); 7.49–7.57 m (5H, C_6_H_5_); 8.34 s (1H, CH=N); 11.00 s (1H, OH). ^13^C-NMR δ (ppm) (DMSO-d_6_): 53.1; 115.3; 117.5; 120.7; 126.3; 128.5; 128.7; 129.5; 129.6; 130.4; 134.7; 142.3; 156.5; 159.2; 160.5; 164.6; 166.5.

*Methyl [2-{[(3-bromo-4-hydroxyphenyl)methylidene]hydrazinylidene}-4-oxo-3-phenyl-1,3-thiazolidin-5-ylidene]acetate (**95**):* Yield 75%, mp = 236–238 °C. ^1^H-NMR δ (ppm) (DMSO-d_6_): 3.83 s (3H, COOCH_3_); 6.81 s (1H, CH=); 7.06 d, 7.67 dd, 7.90 d (3H, 3-Br-4-HO-C_6_**H_3_**, *J* = 8.4, 2.0 Hz); 7.47–7.57 m (5H, C_6_H_5_); 8.34 s (1H, CH=N); 11.06 s (1H, OH). ^13^C-NMR δ (ppm) (DMSO-d_6_): 53.1; 110.3; 115.3; 117.2; 126.7; 128.7; 129.2; 129.5; 129.6; 133.4; 134.7; 142.3; 157.5; 159.0; 160.4; 164.6; 166.5.

*Methyl [3-(2-chlorophenyl)-2-{[(3-ethoxy-4-hydroxyphenyl)methylidene]hydrazinylidene}-4-oxo-1,3-thiazolidin-5-ylidene]acetate (**96**):* Yield 63%, mp = 216–218 °C. ^1^H-NMR δ (ppm) (DMSO-d_6_): 1.36 t (3H, OCH_2_C**H_3_**, *J* = 6.9 Hz); 3.83 s (3H, COOCH_3_); 4.05 q (2H, OC**H_2_**CH_3_, *J* = 6.9 Hz); 6.87 s (1H, CH=); 6.88 d, 7.25 dd, 7.33 d (3H, 3-C_2_H_5_O-4-HO-C_6_**H_3_**, *J* = 8.1, 1.8 Hz); 7.55-7.60 m, 7.70–7.74 m (4H, 2-Cl-C_6_H_4_); 8.30 s (1H, CH=N); 9.75 s (1H, OH). ^13^C-NMR δ (ppm) (DMSO-d_6_): 15.2; 53.2; 64.4; 112.7; 116.1; 116.3; 123.5; 125.2; 129.0; 130.6; 131.5; 131.9; 132.1; 132.2; 141.3; 147.5; 151.1; 157.7; 161.2; 163.8; 166.4.

*Methyl [3-(4-chlorophenyl)-2-{[(2-hydroxyphenyl)methylidene]hydrazinylidene}-4-oxo-1,3-thiazolidin-5-ylidene]acetate (**97**):* Yield 93%, mp = 214–216 °C. ^1^H-NMR δ (ppm) (DMSO-d_6_): 3.84 s (3H, COOCH_3_); 6.87 s (1H, CH=); 6.92–6.96 m, 7.34–7.40 m, 7.56–7.70 m (8H, 2-HO-C_6_**H_4_** and 4-Cl-C_6_**H_4_**); 8.66 s (1H, CH=N); 10.51 s (1H, OH). ^13^C-NMR δ (ppm) (DMSO-d_6_): 53.2; 115.8; 117.0; 119.1; 120.2; 129.7; 130.6; 130.7; 133.4; 133.6; 134.3; 141.7; 158.5; 159.6; 160.2; 164.4; 166.5.

*Methyl [3-(4-chlorophenyl)-2-{[(3-hydroxyphenyl)methylidene]hydrazinylidene}-4-oxo-1,3-thiazolidin-5-ylidene]acetate (**98**):* Yield 90%, mp = 222–224 °C. ^1^H-NMR δ (ppm) (DMSO-d_6_): 3.82 s (3H, COOCH_3_); 6.81 s (1H, CH=); 6.88–6.92 m, 7.17–7.30 m (4H, 2-HO-C_6_**H_4_**); 7.56 d, 7.62 d (4H, 4-Cl-C_6_H_4_, *J* = 8.7 Hz); 8.38 s (1H, CH=N); 9.78 s (1H, OH). ^13^C-NMR δ (ppm) (DMSO-d_6_): 53.1; 113.8; 115.5; 119.4; 120.6; 129.7; 130.5; 130.6; 133.5; 134.2; 135.2; 142.2; 158.1; 160.7; 161.1; 164.5; 166.5.

*Methyl [3-(4-chlorophenyl)-2-{[(4-hydroxyphenyl)methylidene]hydrazinylidene}-4-oxo-1,3-thiazolidin-5-ylidene]acetate (**99**):* Yield 88%, mp = 250–251 °C. ^1^H-NMR δ (ppm) (DMSO-d_6_): 3.82 s (3H, COOCH_3_); 6.80 s (1H, CH=); 6.86 d (2H, 4-HO-C_6_**H_4_**, *J* = 8.7 Hz); 7.55 d (2H, 4-Cl-C_6_**H_4_**, *J* = 8.7 Hz); 7.61–7.66 m (4H, 4-HO-C_6_**H_4_** and 4-Cl-C_6_**H_4_**); 8.33 s (1H, CH=N); 10.19 s (1H, OH). ^13^C-NMR δ (ppm) (DMSO-d_6_): 53.1; 115.1; 116.4; 125.0; 129.7; 130.7; 133.6; 134.2; 142.4; 159.4; 160.4; 161.2; 164.5; 166.5.

*Methyl [3-(4-chlorophenyl)-2-{[(4-hydroxy-3-methoxyphenyl)methylidene]hydrazinylidene}-4-oxo-1,3-thiazolidin-5-ylidene]acetate (**100**):* Yield 84%, mp = 228–230 °C. ^1^H-NMR δ (ppm) (DMSO-d_6_): 3.82 s (3H, OCH_3_); 3.83 s (3H, COOCH_3_); 6.81 s (1H, CH=); 6.87 d, 7.25 dd, 7.35 d (3H, 4-HO-3-CH_3_O-C_6_**H_3_**, *J* = 8.1, 1.8 Hz); 7.56 d, 7.63 d (4H, 4-Cl-C_6_H_4_, *J* = 8.7 Hz); 8.32 s (1H, CH=N); 9.82 s (1H, OH). ^13^C-NMR δ (ppm) (DMSO-d_6_): 53.1; 56.0; 111.3; 115.2; 116.2; 123.4; 125.3; 129.7; 130.7; 133.6; 134.2; 142.4; 148.4; 150.8; 159.2; 160.7; 164.5; 166.5.

*Methyl [3-(4-chlorophenyl)-2-{[(3-ethoxy-4-hydroxyphenyl)methylidene]hydrazinylidene}-4-oxo-1,3-thiazolidin-5-ylidene]acetate (**101**):* Yield %, mp = 221–223 °C. ^1^H-NMR δ (ppm) (DMSO-d_6_): 1.37 t (3H, OCH_2_C**H_3_**, *J* = 6.9 Hz); 3.83 s (3H, COOCH_3_); 4.06 q (2H, OC**H_2_**CH_3_, *J* = 6.9 Hz); 6.81 s (1H, CH=); 6.88 d, 7.25 dd, 7.33 d (3H, 3-C_2_H_5_O-4-HO-C_6_**H_3_**, *J* = 8.1, 1.8 Hz); 7.56 d, 7.63 d (4H, 4-Cl-C_6_H_4_, *J* = 9.0 Hz); 8.30 s (1H, CH=N); 9.73 s (1H, OH). ^13^C-NMR δ (ppm) (DMSO-d_6_): 15.2; 53.1; 64.4; 112.8; 115.2; 116.3; 123.3; 125.3; 129.7; 130.7; 133.6; 134.2; 142.4; 147.5; 151.1; 159.1; 160.8; 164.5; 166.5.

*Methyl [2-{[(3-chloro-4-hydroxyphenyl)methylidene]hydrazinylidene}-3-(4-chlorophenyl)-4-oxo-1,3-thiazolidin-5-ylidene]acetate (**102**):* Yield 85%, mp = 248–250 °C. ^1^H-NMR δ (ppm) (DMSO-d_6_): 3.83 s (3H, COOCH_3_); 6.81 s (1H, CH=); 7.08 d, 7.75 d (2H, 3-Cl-4-HO-C_6_**H_3_**, *J* = 8.4, 2.1 Hz); 7.54-7.64 m (5H, 3-Cl-4-HO-C_6_**H_3_**, and 4-Cl-C_6_**H_4_**); 8.34 s (1H, CH=N); 11.00 s (1H, OH). ^13^C-NMR δ (ppm) (DMSO-d_6_): 53.1; 115.4; 117.5; 120.8; 126.2; 128.6; 129.7; 130.4; 130.7; 133.5; 134.2; 142.3; 156.6; 159.3; 160.3; 164.5; 166.5.

*Methyl [2-{[(3-bromo-4-hydroxyphenyl)methylidene]hydrazinylidene}-3-(4-chlorophenyl)-4-oxo-1,3-thiazolidin-5-ylidene]acetate (**103**):* Yield 88%, mp = 263–264 °C. ^1^H-NMR δ (ppm) (DMSO-d_6_): 3.83 s (3H, COOCH_3_); 6.81 s (1H, CH=); 7.06 d, 7.67 dd, 7.90 d (3H, 3-Br-4-HO-C_6_**H_3_**, *J* = 8.4, 1.8 Hz); 7.55 d, 7.63 d (4H, 4-Cl-C_6_H_4_, *J* = 9.0 Hz); 8.34 s (1H, CH=N); 11.07 s (1H, OH). ^13^C-NMR δ (ppm) (DMSO-d_6_): 53.1; 110.3; 115.3; 117.2; 126.7; 129.2; 129.7; 130.7; 133.4; 133.5; 134.2; 142.3; 157.6; 159.2; 160.3; 164.5; 166.5.

*Methyl [3-(2,4-dichlorophenyl)-2-{[(4-hydroxy-3-methoxyphenyl)methylidene]hydrazinylidene}-4-oxo-1,3-thiazolidin-5-ylidene]acetate (**104**):* Yield 68%, mp = 214–216 °C. ^1^H-NMR δ (ppm) (DMSO-d_6_): 3.82 s (3H, OCH_3_); 3.83 s (3H, COOCH_3_); 6.89 s (1H, CH=); 6.87 d, 7.25 dd, 7.35 d (3H, 4-HO-3-CH_3_O-C_6_**H_3_**, *J* = 8.4, 1.8 Hz); 7.69 dd, 7.78 d, 7.96 d (3H, 2,4diCl-C_6_H_3_, *J* = 8.4, 2.1 Hz); 8.32 s (1H, CH=N); 9.83 s (1H, OH). ^13^C-NMR δ (ppm) (DMSO-d_6_): 53.2; 56.0; 111.3; 116.2; 116.3; 123.6; 125.2; 129.3; 130.3; 131.4; 132.8; 133.3; 135.9; 141.2; 148.4; 150.9; 157.5; 161.3; 163.7; 166.4.

### 3.6. Preparation of Compounds and Drugs

Compounds were dissolved in dimethyl sulfoxide (DMSO, Sigma-Aldrich, St. Louis, MO, USA) to 125 mM. The final concentration of DMSO in the compounds dilutions was not higher than 1.00%. Drug, sulfadiazine [4-amino-N-(2-pyrimidinyl)benzenesulfonamide] (S8626, Sigma-Aldrich) was dissolved in 1M sodium hydroxide (NaOH, Sigma-Aldrich) to 400 mM. Final concertation of NaOH in sulfadiazine dilutions was not higher than 2.5%. Ready solution sulfadiazine and trimethoprim in weight ratio 5:1 (Sul-Tridin 24%, 200 mg/mL [800 mM] + 40 mg/mL [137.78 mM]– ScanVet Poland). Dilution of the all compounds and drugs were freshly prepared before the cells were exposed in appropriate medium.

### 3.7. Cytotoxic Assay

The L929 mouse fibroblast (ATTC^®^ CCL-1™, Manassas, VA, USA) were culturing according to the ATCC product sheet. Cells were trypsinized (Trypsin-EDTA Solution, 1X ATCC^®^ 30-2101™) twice a week and seeded at a density of 1 × 10^6^ per T25 cell culture flask and incubated in a 37 °C and 5% CO_2_ to achieve a confluent monolayer.

The cell viability assay were performed precisely according to international standards (ISO 10993-5:2009(E)), using the tetrazolium salt [3-(4,5-dimethylthiazol-2-yl)-2,5-diphenyltetrazolium bromide] (MTT, Sigma-Aldrich) and mouse fibroblasts L929 cells. Cells (1 × 10^4^ per well) were treated 24 h with compounds or drug in different concentration range. The optical density at 570 nm on the ELISA reader (Multiskan EX, Labsystems, Vienna, VA, USA), was read. The results were expressed as a percentage of viability compared to untreated cells. To calculate the reduction of viability compared to the untreated blank the equation was used: viability (%) = 100% × sample OD_570_ (the mean value of the measured optical density of the treated cells)/blank OD_570_ (the mean value of the measured optical density of the untreated cells). The CC_30_/CC_50_ represents the concentration of tested compounds or drugs that was required for 30% or 50% cells proliferation inhibition in vitro. According to the previously mentioned ISO 10993-5:2009(E) norm, the CC_30_ value is recognized as non-cytotoxic concentration for the cell line. All experiments were performed in triplicate.

### 3.8. Assay In Vitro for Anti-T. gondii Activity

The Hs27 (human foreskin fibroblast) (ATCC^®^ CRL-1634™) were culturing according to the ATCC product sheet. The cell line, when it achieved a confluent monolayer, was trypsinized and seeded at a density of 1 × 10^6^ per T25 cell culture flask and incubated for 48–72 h in a 37 °C and with 5% CO_2_. The RH strain of *Toxoplasma gondii* (ATCC^®^ PRA-310™) is highly virulent were maintained as tachyzoites according to the ATCC product sheet. Infected tissue culture cells were incubated in a 37 °C and 5% CO_2_. 

The influence of compounds and drugs on *T. gondii* proliferation was described in our previous work [20,21,22]. Due to the fact that *T. gondii* is an intracellular parasite and host cells should not be destroyed by the cytotoxic concentration of compounds, to determine the initial dose of each compound to antiparasitic study CC_30_ value was calculated (data not shown). In our study we use a [^3^H]-uracil incorporation assay, which is specific for the labelling of the nucleic acids of the parasite. Shortly, 1×10^4^ per well of Hs27 cells were seeded on 96-well plates. After 72 h of incubation, tachyzoites of the RH strain were added to the cell monolayers, ratio 1:10, and incubate for 1h. Then compounds or drug in different concentration range, were added to the Hs27 cells with *T. gondii*. After a subsequent 24 h of incubation, 1 µCi/well [5,6-^3^H] uracil (Moravek Biochemicals Inc., Brea, CA, USA) was added to each microculture for a further 72 h. The amount of the isotope incorporated into the parasite nucleic acid pool, corresponding to the parasite growth, was measured by liquid scintillation counting using 1450 Microbeta Plus Liquid Scintillation Counter (Wallac Oy, Turku, Finland). The results were expressed as counts per minute (CPM) and transformed to the percentage of viability compared to untreated cells. The IC_50_ represents the concentration of tested compounds or drugs that was required for 50% inhibition of *T. gondii* proliferation in vitro. All experiments were performed in triplicate.

### 3.9. Graphs and Statistical Analyses

Statistical analyses and graphs were performed using GraphPad Prism version 8.0.1 for macOS (GraphPad Software, San Diego, CA, USA). For compounds with CC_30_/CC_50_ or IC_50_ values greater than the highest concentration tested, values were calculated based on extrapolation of the curves using the GraphPad Prism program (version 8.0.1). The standard deviation (SD) were calculated from a sample of values, also using the GraphPad Prism. Selectivity ratio (SR) values were calculated as the ratio of the 50% cytotoxic concentration (CC_50_) to the 50% antiparasitic concentration (IC_50_).

## 4. Conclusions

We evaluated a series of new thiazolidin-4-one derivatives for their ability to inhibit in vitro growth of the parasite *T. gondii*. All active compounds inhibit proliferation of *T. gondii* tachyzoites better than used references drugs both sulfadiazine as well as synergistic effect of sulfadiazine + trimethoprim (5:1). The most active of them (compounds **94** and **95**) showed anti-*T. gondii* activity over 392-fold higher than sulfadiazine and 18-fold higher than sulfadiazine + trimethoprim (5:1). All active compounds (**82**–**88** and **91**–**95**) against *T. gondii* represent selectivity ratio values from 1.75 to 15.86 (CC_30_/IC_50_) which shows that were more towards inhibiting parasite proliferation vs. cytotoxicity.

The results of these studies also collectively suggest that thiazolidin-4-ones containing in their structure halogenphenylthiosemicarbazide moiety was useful for their antitoxoplasmic activity that confirmed their inhibitory potential against *T. gondii* tachyzoites. Received funding from the National Science Centre allow (project: Antiparasitic properties and molecular target identification of new thiosemicarbazide and thiazolidinone derivates in *Toxoplasma gondii* invasions) to determine a molecular target(s) for selected new compounds from thiosemicarbazide and thiazolidinone derivatives possessing the most potent anti-*T. gondii* activity.

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
