# Peer review of "Synthesis and In Vitro Anti-Toxoplasma gondii Activity of Novel Thiazolidin-4-one Derivatives"

_molecules, 2019, doi:10.3390/molecules24173029_

Round 1

Reviewer 1 Report

Review of Trotsko et al., “Synthesis and in vitro anti-Toxoplasma gondii activity of novel thiazolidine-4-one derivatives

In this manuscript, the authors describe the synthesis of a series of thiazolidine-4-one derivatives with activity against the apicomplexan parasite Toxoplasma gondii.  Compounds were developed that were active against intracellular tachyzoites, with EC50values in the 10-100’s of uM range.  This work expands our understanding of the anti-parasitic activity of thiazolidine-4-one derivatives, which have been found by this group and others to have relatively broad-spectrum activity against protozoan parasites.  As a consequence, this manuscript may be of interest to those in the medicinal chemistry field, particularly those interested in development of therapies against T. gondii.

Comments:

1.     Throughout, the insufficient statistical analysis is used to discuss similarities or differences in compound activities. Without some idea of the SD of the CC50and IC50values in Tables 1 and 2, it is difficult to determine if differences in values are real. 

Related to this, it is not clear how satisfactory CC50 values was established (see line 134).  For example, the compound with a CC50value of 484.97 uM may be less toxic than one with a value of 385.61 uM (depending on SD, which needs to be included), but this difference is arbitrary. Similarly, one cannot say that “All active compounds were non-toxic in cytotoxicity tests…” (line 24-25, abstract), as the compounds did have a solvable CC50value.  This is also true on line 144 (“derivatives showed no cytotoxicity (CC50=427.98-492.91)”.  Clearly the compounds were toxic at some level.  

2.     The IC50values of some compounds vs. parasites is 3-4-fold the CC50values of the compounds.  How is the death of host cells in these cases included in the calculation, as their death would impact parasite health?  Additionally, please provide a column showing the selectivity index in Table 2.

3.     The review will need extensive editing to correct the English.  (Examples are included below, but these are by no means exhaustive):

a.    Intro – line 30, “invaded this parasite” to “are infected by this parasite” 

b.    Intro line 35, remove “;schizophrenia, Parkinsons’s disease or epilepsy, due to…” replace with ““,schizophrenia, Parkinsons’s disease or epilepsy.  Remove the reset of the sentence. 

c.    Line 42, remove “In turn”, start sentence “In immunocompromised persons…”

d.    Line 42, “among others headaches and around the chest” is missing something. 

e.    Line 46, “what’s” is too casual.

f.     Line 51, replace clause with “the interest in thiazolidine-4-one derivatives has increased”

g.    Line 66, Line 42, remove “In turn”, start sentence “The results..”

h.     The table formats are disrupted, making them unclear.  

Author Response

Response to comments

Detailed responses (marked in blue) to reviewers’ comments

Reviewer1

Open Review

Comments and Suggestions for Authors

Review of Trotsko et al., “Synthesis and in vitro anti-Toxoplasma gondii activity of novel thiazolidine-4-one derivatives

In this manuscript, the authors describe the synthesis of a series of thiazolidine-4-one derivatives with activity against the apicomplexan parasite Toxoplasma gondii.  Compounds were developed that were active against intracellular tachyzoites, with EC50values in the 10-100’s of uM range.  This work expands our understanding of the anti-parasitic activity of thiazolidine-4-one derivatives, which have been found by this group and others to have relatively broad-spectrum activity against protozoan parasites.  As a consequence, this manuscript may be of interest to those in the medicinal chemistry field, particularly those interested in development of therapies against T. gondii.

 Comments:

Throughout, the insufficient statistical analysis is used to discuss similarities or differences in compound activities. Without some idea of the SD of the CC50 and IC50 values in Tables 1 and 2, it is difficult to determine if differences in values are real. Related to this, it is not clear how satisfactory CC50 values was established (see line 134).  For example, the compound with a CC50 value of 484.97 uM may be less toxic than one with a value of 385.61 uM (depending on SD, which needs to be included), but this difference is arbitrary.

 It has been corrected as suggested. Missing SD values (Table 1, Table 2 and line 137, 139, 148-150, 171-173, 189, 199-201, and 204-205) were added.

Similarly, one cannot say that “All active compounds were non-toxic in cytotoxicity tests…” (line 24-25, abstract), as the compounds did have a solvable CC50value.  This is also true on line 144 (“derivatives showed no cytotoxicity (CC50=427.98-492.91)”.  Clearly the compounds were toxic at some level.  

 It has been corrected as suggested:

line 24-25, abstract - now line: 24-26: “All active compounds (82-88 and 91-95) against T. gondii represent values from 1.75 to 15.86 (CC30/IC50) lower than no cytotoxic value (CC30).” line 144 – now line 149-152: ” The remaining (4-oxothiazolidin-5-ylidene)acetic acid derivatives showed not high cytotoxicity (CC50=427.98 ± 21.90 - 492.91 ± 19.89 µM) and were no cytotoxic at the similar level (CC30=315.97.98 ± 20.45 – 471.94 ± 20.20 µM, data not shown).” The IC50values of some compounds vs. parasites is 3-4-fold the CC50values of the compounds.  How is the death of host cells in these cases included in the calculation, as their death would impact parasite health?  Additionally, please provide a column showing the selectivity index in Table 2.

As we mention in the article in first step the cytotoxic study was performed and CC30 (line 127-133) values were established. On these base the initial dose of each compound to anti-Tg study, was determined, to excluded death cell in calculation, but data was not shown.

Compound

CC50
[µM]

CC30
[µM]

IC50
[µM]

SDIC50
[µM]

SR
[CC50/IC50]

SR
[CC30/IC50]

75

≥1000

947.50

na

76

≥1000

541.43

na

77

≥1000

≥1000

na

78

≥1000

778,82

na

79

≥1000

≥1000

na

80

≥1000

707.13

na

82

656.20

485.34

263.22

± 19.31

2.49

1.84

83

906.30

713.16

271.15

± 24.96

3.34

2.63

84

827.45

746.78

219.93

± 19.22

3.76

3.39

85

759.22

651.94

165.40

± 16.02

4.59

3.94

86

≥1000

465.06

177.72

± 18.59

≥5.63

2.62

87

424.66

319.01

115.92

± 21.68

3.66

2.75

88

565.86

437.74

129.42

± 14.14

4.37

3.38

91

492.91

471.94

92.88

± 14.94

5.31

5.08

92

486.11

315.97

180.27

± 21.91

2.69

1.75

93

463.03

448.93

191.21

± 27.32

2.42

2.35

94

473.73

440.06

27.74

± 4.27

17.07

15.86

95

427.98

404.08

28.32

± 5.83

15.11

14.27

For example: compound 94 has CC30 = 440.06±15.07 µM so initial (highest) dose in antiparasitic study was on level 440 µM and them serially diluted. In graphs the IC50 curves were extended to 500 µM using GraphPad Prism, if the initial dose was lower than this value.

Furthermore, editor ask us “Why CC30 instead CC50 have been used? This latter would allow a better comparison of selectivity”. In the answer to the question we changed values from CC30 to CC50. ​

The review will need extensive editing to correct the English. (Examples are included below, but these are by no means exhaustive): Intro – line 30, “invaded this parasite” to “are infected by this parasite”  Intro line 35, remove “;schizophrenia, Parkinsons’s disease or epilepsy, due to…” replace with ““,schizophrenia, Parkinsons’s disease or epilepsy.  Remove the reset of the sentence.  Line 42, remove “In turn”, start sentence “In immunocompromised persons…” Line 42, “among others headaches and around the chest” is missing something.  Line 46, “what’s” is too casual. Line 51, replace clause with “the interest in thiazolidine-4-one derivatives has increased” Line 66, Line 42, remove “In turn”, start sentence “The results..” The table formats are disrupted, making them unclear.  

It has been corrected as suggested. The manuscript has been edited by an English-speaking native. The Table formats have been improved.

Reviewer 2 Report

The authors of this manuscript investigated the anti-Toxoplasma activity of Thiazolidin-4- derivatives. 

The investigation of the novel Thiazolidin effect on parasite  T. gondi is attractive research. However, the reviewer has some concerns.

Comments

1. Toxoplasma parasite invades in various tissue systems, authors have tried only the fibroblast cell model. It may be a good option to try a specific cell model to check the effect of the drug on parasite sustainability.

2. Authors should discuss the mechanism involved in parasite inhibitory pathways.

3. These compounds are not soluble in water. DMSO is cytotoxic in nature. If the drugs are prepared in DMSO how safe to treat in-vivo.

4. Method of thiosemicarbazones are too detailed, it could be explained in brief.

Author Response

Response to comments

Detailed responses (marked in blue) to reviewers’ comments

Reviewer2

The authors of this manuscript investigated the anti-Toxoplasma activity of Thiazolidin-4- derivatives. 

The investigation of the novel Thiazolidin effect on parasite  T. gondi is attractive research. However, the reviewer has some concerns.

Comments

Toxoplasma parasite invades in various tissue systems, authors have tried only the fibroblast cell model. It may be a good option to try a specific cell model to check the effect of the drug on parasite sustainability.

The use of Hs27 cell line and T. gondii RH strain is widely distributed and very popular in many laboratories when testing new compounds. This approach allows for direct comparison of results between different research centres. Furthermore, Hs27 cell line is dedicated for the T. gondii proliferation according to

https://www.lgcstandards-atcc.org/products/all/PRA-310.aspx#culturemethod

Growth Conditions

Temperature: 35°C to 37°C

Cell Line: ATCC® CRL-1634™ (human foreskin fibroblasts)

Alternate Cell Line: ATCC® CCL-81™ (African green monkey kidney)”.

 In addition, our experience shows that the use of different continuous cell lines does not have significant differences in the proliferation of Toxoplasma tachyzoites because they can  invade any nucleated cell. Moreover the in vitro study are the first step of the selection of most active anti-Tg compound and next used them in in vivo study due to, as you mention above, its invades in various tissue systems and especially has tissue tropism to the nervous system.

Authors should discuss the mechanism involved in parasite inhibitory pathways.

In our study we showed that our novel thiazolidinone derivatives inhibited the parasite replication in vitro. Twelve of them stood out as promising antiparasitic agents possessing an activity better than sulfadiazine or its combination with trimethoprim against Toxoplasma and lower cytotoxicity with respect to the reference drug. Unfortunately, no one has yet explained the mechanism of action of thiazolidinone derivatives in T. gondii and their molecular target(s) in this parasite remain unknown. But, recently we have received funding from the National Science Centre for the project whose the main goal is to determine a molecular target(s) for selected new compounds from thiosemicarbazide and thiazolidinone derivatives which possess the most potent anti-T. gondii activity.

These compounds are not soluble in water. DMSO is cytotoxic in nature. If the drugs are prepared in DMSO how safe to treat in-vivo.

Firstly we check the cytotoxicity of DMSO on L292 and Hs27 lines. In our study we treated both cell lines with DMSO in concentration range 4.0–0.03% and no cytotoxic effect was observed. Also in our previously study we use DMSO as a compounds solvent.

Paneth, A.; Węglińska, L.; Bekier, A.; Stefaniszyn, E.; Wujec, M.; Trotsko, N.; Dzitko, K. Systematic identification of thiosemicarbazides for inhibition of Toxoplasma gondii growth in vitro. Molecules 2019, 24, 614. Paneth, A.; Węglińska, L.; Bekier, A.; Stefaniszyn, E.; Wujec, M.; Trotsko, N.; Hawrył, A.; Hawrył, M.; Dzitko, K. Discovery of potent and selective halogen-substituted imidazole-thiosemicarbazides for inhibition of Toxoplasma gondii growth in vitro via structure-based design. Molecules 2019, 24, 1618.

Furthermore, to in vitro study we dilute compounds stock (125 mM) at least 250 times. The most active compounds (94 and 95) showed anti-T. gondii activity in concentration aprox. 28 µM, so the concentration of DMSO in both IC50 values were just 0.2 %.

Moreover, according to Institutional Animal Care and Use Committees recommendation, DMSO can be used as a compound solvent in dose 0.5% – 5%. Due to its anti-inflammatory properties and the ability to scavenge reactive oxygen particles, DMSO has been purposed for the treatment of several diseases. Also some drugs that are not soluble in water are often soluble in DMSO.

Xing, L and Remick, D.G. Mechanisms of Dimethyl Sulfoxide Augmentation of IL-1β Production. The Journal of Immunology, 2005, 174:6195-6202. Brayton, C.F. Dimethyl sulfoxide (DMSO): a review. Cornell Vet. 1986 Jan: 76(1):61-90.

Method of thiosemicarbazones are too detailed, it could be explained in brief.

 The method of thiosemicarbazones synthesis was described according to guidelines for Molecules. Such presentation of results provides clearly the source of physicochemical properties of previously described compounds.

Round 2

Reviewer 1 Report

This resubmitted manuscript from Trotsko et al., "Synthesis and in vitro anti-Toxoplasma gondii activity of novel thiazolidin-4-one derivatives" describes the development and testing of a class of bioactive compounds against an important parasite. 

Most of the comments in the original review were adequately addressed.  There is still too much confusion created by English grammar problems, and I have concerns about the editing that was purported to be performed by a native English speaker.

For example:

Line 42 (same as original paper) - "In immunocompromised persons...acute phase of infectious."  This sentence needs revising for clarity.  Line 149 - "showed not high" - limited?   Line 148 - "The remaining...data not shown)." - this entire thought is unclear and needs to be revised for clarity. Please italicize genus and species throughout (line 25, for example).